# Concentrated solar $CO_2$ reduction in $H_2O$ vapour with >1% energy conversion efficiency

Yuqi Ren[1,5], Yiwei Fu[2,5], Naixu Li[1] ✉, Changjun You[1], Jie Huang[2], Kai Huang[1], Zhenkun Sun[3], Jiancheng Zhou[1], Yitao Si[1], Yuanhao Zhu[1], Wenshuai Chen[4], Lunbo Duan[3] ✉ & Maochang Liu[2] ✉

$H_2O$ dissociation plays a crucial role in solar-driven catalytic $CO_2$ methanation, demanding high temperature even for solar-to-chemical conversion efficiencies <1% with modest product selectivity. Herein, we report an oxygen-vacancy ($V_o$) rich $CeO_2$ catalyst with single-atom Ni anchored around its surface $V_o$ sites by replacing Ce atoms to promote $H_2O$ dissociation and achieve effective photothermal $CO_2$ reduction under concentrated light irradiation. The high photon flux reduces the apparent activation energy for $CH_4$ production and prevents $V_o$ from depletion. The defects coordinated with single-atom Ni, significantly promote the capture of charges and local phonons at the Ni $d$-impurity orbitals, thereby inducing more effective $H_2O$ activation. The catalyst presents a $CH_4$ yield of 192.75 μmol/cm$^2$/h, with a solar-to-chemical efficiency of 1.14% and a selectivity ~100%. The mechanistic insights uncovered in this study should help further the development of $H_2O$-activating catalysts for $CO_2$ reduction and thereby expedite the practical utilization of solar-to-chemical technologies.

$CO_2$ reduction involves multiple proton-coupled electron transfers, where the final product is determined based on the kinetic and thermodynamic parameters of the reduction pathway[1–4]. $CH_4$ formation is thermodynamically more favourable than $CO_2$ reduction to CO, but it is kinetically more challenging[5–7]. Although using $H_2O$ is the most cost-effective proton source, $H_2O$ dissociation is the primary rate-limiting step in the photo-thermal catalytic $CH_4$ production from $CO_2$ and $H_2O$[8–10], requiring high temperatures for very modest (<1%) energy conversion efficiencies[11]. To circumvent these issues, efficient activation of $H_2O$ molecules, separation of photocatalysis and thermo-catalysis on a microscopic scale, and generation of active sites that can capture localised phonons as well as photo-generated charge carriers are key research goals in catalyst development.

Single-atom Ni is widely used as a catalyst for the oxygen evolution reaction to promote the dissociation of $H_2O$ and production of reactive oxygen species while enhancing charge trapping and transfer[12–15]. In addition, single-atom Ni can generate hot spots with strong electron–phonon coupling, which overlap with the captured photo-generated charge carriers[16–18], thereby enhancing the molecular dissociation at the catalyst surface and improving the solar-to-chemical (STC) energy conversion efficiency. Accordingly, we set out to investigate a single-atom Ni-based catalyst approach for the efficient photo-thermal reduction of $CO_2$ with $H_2O$ to $CH_4$ under concentrated solar irradiation conditions.

The catalyst, in the form of single-atom Ni-loaded oxygen-vacancy ($V_o$) rich $CeO_2$, presented a $CH_4$ yield of 192.75 μmol/cm$^2$/h with the selectivity of ~100%. Significantly, the STC energy conversion

[1]School of Chemistry and Chemical Engineering, Southeast University, No. 2 Dongnandaxue Road, Nanjing, Jiangsu 211189, PR China. [2]International Research Center for Renewable Energy, State Key Laboratory of Multiphase Flow in Power Engineering, Xi'an Jiaotong University, No.28 Xianning West Road, Xi'an, Shaanxi 710049, PR China. [3]Key Laboratory of Energy Thermal Conversion and Control of Ministry of Education, School of Energy and Environment, No. 2 Dongnandaxue Road, Nanjing, Jiangsu 210096, PR China. [4]Key Laboratory of Bio-Based Material Science and Technology, Ministry of Education, Northeast Forestry University, No. 26 Hexing Road, Harbin 150040, PR China. [5]These authors contributed equally: Yuqi Ren, Yiwei Fu. ✉e-mail: naixuli@seu.edu.cn; duanlunbo@seu.edu.cn; maochangliu@mail.xjtu.edu.cn

efficiency in this system reaches 1.14%. In-depth mechanism investigations are performed using both experimental and theoretical methods. Our findings demonstrate that $V_o$ regenerated on the catalyst surface ensures a steady supply of active sites for $CO_2$ reduction. Density functional theory (DFT) and time-dependent DFT (TDDFT) calculations confirm the crucial roles of Ni and $V_o$ in enhancing carrier kinetics and reactant activation. Furthermore, ab initio molecular dynamics (AIMD) and TDDFT simulations demonstrate that the $H_2O$ dissociation on the catalyst surface proceed via thermally assisted photo-catalysis. In this work, we offer an effective photo-thermal approach to achieve high STC energy conversion by integrating concentrating solar irradiation and elucidates how the surface atomic structures impact the catalytic efficiency.

## Results

### Structural characterization of catalysts

$CeO_2$ exhibits a strong response to the full spectrum, high catalytic stability under high temperatures and pressures[19–21], and abundant $V_o$ and lattice defects on its surface[22–24]. Based on these characteristics, we initially fabricated $CeO_2$ non-porous nanorods using the hydrothermal method with strict temperature control and chemical reduction, which allowed for creating multiple $V_o$ and anchoring single-atom Ni centres on their surface (Supplementary Fig. 1). The X-ray diffraction (XRD) pattern of the obtained products (Supplementary Fig. 2) can be accurately indexed to the standard fluorite structure of $CeO_2$ (PDF card no. 43-1002). No notable diffraction peaks for Ni were observed, probably because of the low Ni content. Transmission electron microscopy (TEM) images revealed that NF@0.1%Ni@CeO$_2$-$V_o$ and NF@0.1%Ni@CeO$_2$ exhibited a porous nanorod morphology (Fig. 1a and Supplementary Figs. 3a, b), which increases the catalyst-specific surface area, enhancing the $CO_2$ adsorption and providing more active sites (Supplementary Fig. 4; Supplementary Table 1). In-situ CO-DRIFT images failed to identify whether Ni is a single atom due to the poor chemisorption of CO on the non-precious metal Ni (Supplementary Figs. 5, 6). HAADF-STEM images of NF@0.1%Ni@CeO$_2$-$V_o$ show lattice stripes with a crystal plane spacing of 0.31 nm, corresponding to the (111) crystal plane of $CeO_2$ on the primarily exposed planes (Fig. 1b). Besides, the presence of Ni single atoms was observed (Fig. 1c).

The energy-dispersive X-ray spectroscopy elemental mapping images (Fig. 1d–g) showed that Ce, O and Ni were evenly distributed. The electronic and coordinative structures of NF@0.1%Ni@CeO$_2$-$V_o$ and NF@0.1%Ni@CeO$_2$ were supported by X-ray absorption spectroscopy (XAS) results. According to the X-ray absorption near-edge spectra at the Ni K-edge of NF@0.1%Ni@CeO$_2$-$V_o$ and NF@0.1% Ni@CeO$_2$ (Fig. 1h), the $E_0$ of NF@0.1%Ni@CeO$_2$-$V_o$ is similar to that of NiO, suggesting that Ni atoms carry a +2 charge, while the $E_0$ of NF@0.1%Ni@CeO$_2$ is similar to that of $Ni_2O_3$, indicating an oxidation state of +3 for the Ni atoms. This can be attributed to the presence of $V_o$, which enables the transfer of electrons from $O^{2-}$ on the catalyst surface to Ni, reducing $Ni^{3+}$ to the +2 valence state; conversely, Ni remains in the +3 valence state in the absence of $V_o$ (Supplementary Fig. 7).

The coordination environment of single-atom Ni was determined using extended X-ray absorption fine-structure (EXAFS) spectroscopy. The Fourier-transformed $k^2$-weighted EXAFS spectra at the Ni K-edge of NF@0.1%Ni@CeO$_2$-$V_o$ and NF@0.1%Ni@CeO$_2$ (Fig. 1i) show a prominent peak at the position of the Ni–O shell (1.5 Å), which is absent in the spectra of the reference samples of Ni foil and NiO. The EXAFS data fitting results (Supplementary Table 2) indicate that the coordination number of Ni–O in NF@0.1%Ni@CeO$_2$-$V_o$ is 4.3, whereas that of Ni–O in NF@0.1%Ni@CeO$_2$ is 5.1 (Supplementary Fig. 8a–d). These results suggest that the introduction of $V_o$ causes surface reconstruction, with Ni replacing Ce and anchored around the surface $V_o$ sites, decreasing the $H_2O$ dissociation energy The coordination environment of Ni–O in NF@0.1%Ni@CeO$_2$-$V_o$ and NF@0.1%Ni@CeO$_2$ was modelled using

density functional theory (DFT) (Fig. 1j, k), revealing that the single-atom Ni loaded on the (111) face replaced Ce, forming two structures with coordination numbers and bond lengths that align with the XAS results (Supplementary Table 3). The Ni foil exhibits a wave flap associated with Ni–Ni coordination at (7.2 Å, 2.2 Å$^{-1}$) (Fig. 1l), whereas NiO and $Ni_2O_3$ present an additional wave flap at (5.5 Å, 1.5 Å$^{-1}$), which is attributed to Ni–O coordination (Fig. 1m, n). In the case of NF@0.1% Ni@CeO$_2$-$V_o$ and NF@0.1%Ni@CeO$_2$, the absence of wave flaps at high k values (Fig. 1o, p) indicates that the central Ni is not bound to another heavy atom; in contrast, wave flaps appear at low k values (5.5 Å, 1.5 Å$^{-1}$), which can be attributed to Ni–O coordination. Meanwhile, XAS and EPR analyses confirmed the electronic structure, coordination structure of the Ce, and $V_o$ content of the catalysts (Supplementary Figs. 9, 10; Supplementary Table 4).

### Photo-thermal catalytic performance

The presence of $V_o$ can exert diverse effects on the local coordination environments of Ni (i.e., $NiO_4$ and $NiO_5$). To explore the impact of different coordination environments on the surface temperature and photo-thermal catalytic performance, we utilised the reaction between $CO_2$ and $H_2O$ to form $CH_4$ under concentrated solar irradiation as a probe. The infrared (IR) images presented in Supplementary Fig. 11 indicate the surface temperatures of NF@0.1%Ni@CeO$_2$-$V_o$ and NF@0.1%Ni@CeO$_2$ under concentrated solar irradiation (those recorded under non-concentrated solar irradiation conditions are presented in Supplementary Fig. 12). Meanwhile, light source spectra are provided in Supplementary Fig. 13. NF@0.1%Ni@CeO$_2$-$V_o$ stabilised at 362.1 °C in 7.5 s, while NF@0.1%Ni@CeO$_2$ stabilised at 306.5 °C in 12.6 s (Fig. 2a). This difference in surface temperature can be attributed to Ni forming a local octahedral coordination with lattice O atoms in the absence of $V_o$, resulting in the formation of non-occupied mid-gap states. These states can trap hot electrons and, in combination with holes, can locally exhibit a hot-spot effect (Fig. 2b). In contrast, in the presence of $V_o$, Ni forms a planar quadrilateral coordination closer to the valence-band maximum (VBM), which can trap holes that subsequently recombine with electrons at the conduction-band minimum (CBM), generating a hot-spot effect (Fig. 2c). However, the impurity state of the $d$ orbital is located at the top of the VBM, which implies that the VBM hole capture is equivalent to a relaxation process with an energy level that is similar to the bandgap value between the CBM and VBM, increasing the possibility of capturing the photo-generated holes (Detailed description in Supplementary Fig. 14). The wave function distribution of the corresponding mid-gap states confirms this hypothesis (Supplementary Fig. 15). Moreover, the electron–phonon interactions in NF@0.1%Ni@CeO$_2$ (Fig. 2b) include two steps of scattering to form two phonons of low frequencies. However, in NF@0.1% Ni@CeO$_2$-$V_o$ (Fig. 2c), only one phonon of a high frequency is formed, resulting in a higher average surface temperature. Besides, the scattering illustrated in Fig. 2b is a multi-stage process, resulting in a lower scattering probability.

To gain insight into the catalytic $CO_2$ reduction process under concentrated solar irradiation conditions, we evaluated the thermocatalysed $CO_2$ reduction using NF@0.1%Ni@CeO$_2$-$V_o$ under dark conditions. Upon increasing the temperature from 200 °C to 400 °C, no $CH_4$ was produced (Fig. 2d). Conversely, under simulated solar light irradiation of 1200 mW/cm$^2$, the $CH_4$ yield substantially increased. In addition, catalytic experiments were conducted with cooled circulating water or under irradiation of variable wavelengths (Supplementary Figs. 16, 17, 18). The catalytic reaction is facilitated by raising the temperature and photogenerated electrons play a key role. The above results suggest that the $CO_2$ photo-thermal catalysis is proceeded in a manner of thermally assisted photo-catalysis. Then, we used plastic Fresnel lenses as cost-effective solar concentrators to perform experiments under dark, non-concentrated solar irradiation and concentrated solar irradiation conditions (Fig. 2e). The concentrated solar

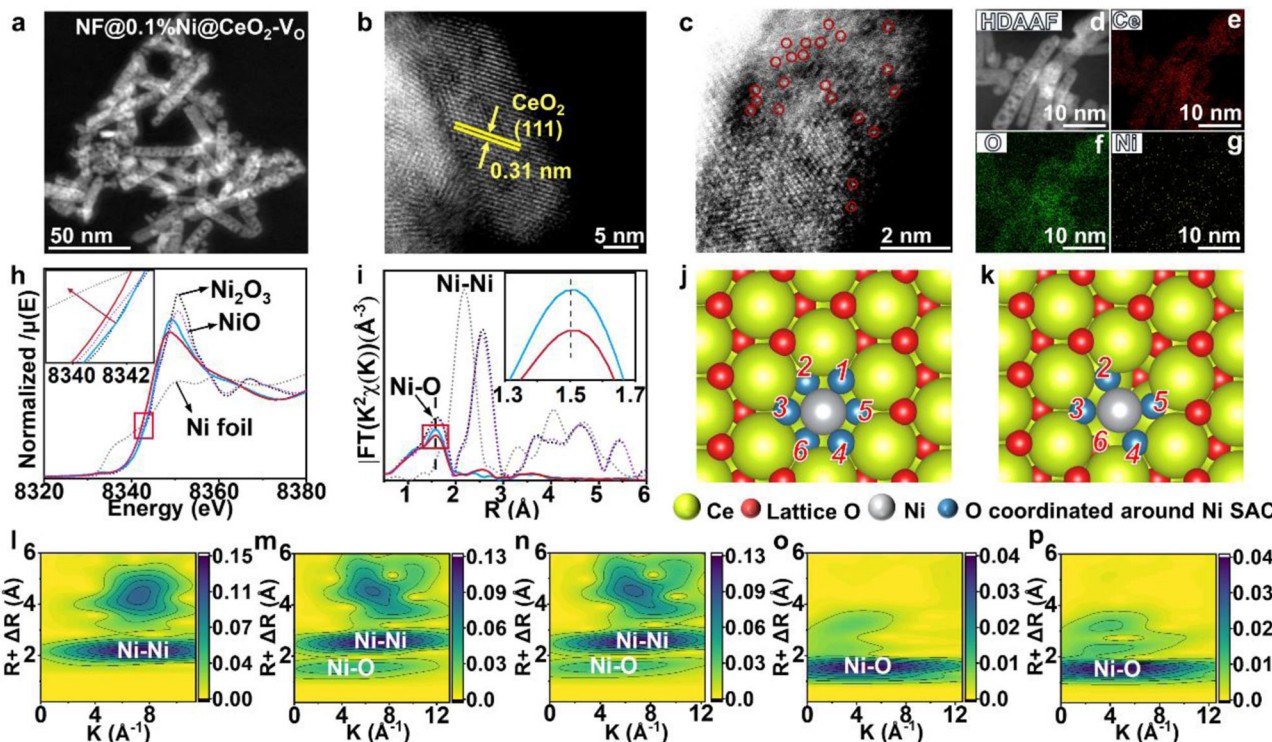

Fig. 1 | Morphological characterizations of NF@0.1%Ni@CeO$_2$-V$_o$. a TEM image of NF@0.1%Ni@CeO$_2$-V$_o$. b, c HAADF-STEM image of NF@0.1%Ni@CeO$_2$-V$_o$. d–g HDAAF (d) EDX elemental mapping images of Ce (red) (e), O (green) (f), and Ni (yellow) (g) in NF@0.1%Ni@CeO$_2$-V$_o$. h, i Normalized X-ray absorption near-edge spectra at the Ni K-edge (h) and k$^2$-weighted Fourier transform extended X-ray absorption fine-structure spectra (EXAFS) in r-space (i) for NF@0.1%Ni@CeO$_2$ (blue curve) and NF@0.1%Ni@CeO$_2$-V$_o$ (red curve) catalysts. j, k Ni−O coordination environment in NF@0.1%Ni@CeO$_2$ (j) and NF@0.1%Ni@CeO$_2$-V$_o$ (k) catalysts. l–p Wavelet Transformation for the k$^2$-weighted EXAFS signal of the Ni foil (l), NiO (m), Ni$_2$O$_3$ (n), NF@0.1%Ni@CeO$_2$ (o), and NF@0.1%Ni@CeO$_2$-V$_o$ (p) catalysts.

irradiation reactor and specific experimental parameters are described in detail in Supplementary Fig. 19a, b. Meanwhile, with conventional lenses, Fresnel lenses can focus parallel light rays to a closer common focal length and reduce the absorption of light owing to their thin thickness (Supplementary Fig. 19c)[25]. NF@0.1%Ni@CeO$_2$-V$_o$ afforded a CH$_4$ yield of 192.75 μmol/cm$^2$/h under concentrated solar irradiation conditions, which was 78 times higher than that achieved under non-concentrated solar irradiation (2.47 μmol/cm$^2$/h) and 42 times higher than that obtained with NF@0.1%Ni@CeO$_2$ under concentrated solar irradiation conditions (4.62 μmol/cm$^2$/h). Additionally, the optimal catalyst preparation conditions were screened (Supplementary Figs. 20, 21). The concentration of V$_o$ can be regulated by the hydro-thermal temperature (Supplementary Fig. 22). Under identical conditions, catalytic activities of 0.1%Ni@CeO$_2$-V$_o$ catalysts loaded on nickel foam did not present notable difference from those of 0.1%Ni@CeO$_2$-V$_o$ powder catalysts (Supplementary Fig. 23a, b). This result demonstrates that nickel foam does not actively participate in the reaction. Therefore, it is rational to consider nickel foam as a support material for the catalyst. To identify the carbon source of the reduced products, isotope tracing experiments were carried out using reactant $^{13}$CO$_2$. The main signal at m/z = 17 is assigned to $^{13}$CH$_4$ and the other signal at m/z = 29 can be assigned to $^{13}$CO. This result indicates that the CO and CH$_4$ generation originates from the CO$_2$ reactant, thereby ruling out catalyst decomposition or carbon pollution as contributing factors (Supplementary Fig. 24).

The energy conversion efficiency is a crucial indicator for evaluating photocatalytic activity. Three benchmark photo-thermal catalysts were synthesized for comparison with the present catalyst and existing studies (WO$_3$, ZrO$_2$, and TiO$_2$, see Supplementary Figs. 25, 26 for the structure/phase information). The direct STC energy conversion of the present catalyst exhibits high efficiency (1.14%; Fig. 2f;

Supplementary Fig. 27; Supplementary Table 5), with a CH$_4$ selectivity of approximately 100% (Supplementary Fig. 28; Supplementary Table 6). Moreover, the differential thermal analysis and thermo-gravimetric analysis results indicate that NF@0.1%Ni@CeO$_2$-V$_o$ has high-temperature resistance (Supplementary Fig. 29a, b). According to the XAS of NF@0.1%Ni@CeO$_2$-V$_o$ after the CO$_2$ reduction reaction under concentrated solar irradiation conditions (Supplementary Figs. 30, 31; Supplementary Tables 7, 8), the valence state and coordination environment of Ni in the catalyst did not clearly change, confirming the strong thermal stability of the catalyst. The catalytic performance did not decline significantly after six cycles and storage in an Ar atmosphere for 30 days (Supplementary Fig. 32), indicating high catalytic stability. To elucidate the reason for the high catalytic activity under concentrated solar irradiation conditions, we subjected NF@0.1%Ni@CeO$_2$ and NF@0.1%Ni@CeO$_2$-V$_o$ to UV−vis diffuse reflectance spectroscopy (DRS). An enhanced full-spectrum absorption response was observed for NF@0.1%Ni@CeO$_2$-V$_o$, as evidenced by a convex peak attributable to the V$_o$ absorption near 1400 and 1800 nm (Supplementary Fig. 33)[26]. This suggests that the V$_o$ may absorb a portion of the energy from the IR light, thereby enhancing the thermal effect in the photo-thermal coupling process. Furthermore, we conducted in-situ XRD tests, finding that the catalyst diffraction peak shifted towards the small-angle direction under concentrated solar irradiation. This phenomenon can be attributed to two reasons: (1) thermal expansion and (2) high flux photon density. The variable-temperature XRD patterns exhibit no shift of the diffraction peaks with increasing temperature. In fact, thermal expansion induces uniform lattice distortions and the distance changes between each crystal plane. However, the changes in lattice constants induced by thermal effects are limited, so the shifts in the positions of the diffraction peaks are weak (Supplementary Fig. 34). On the other hand, high flux photon

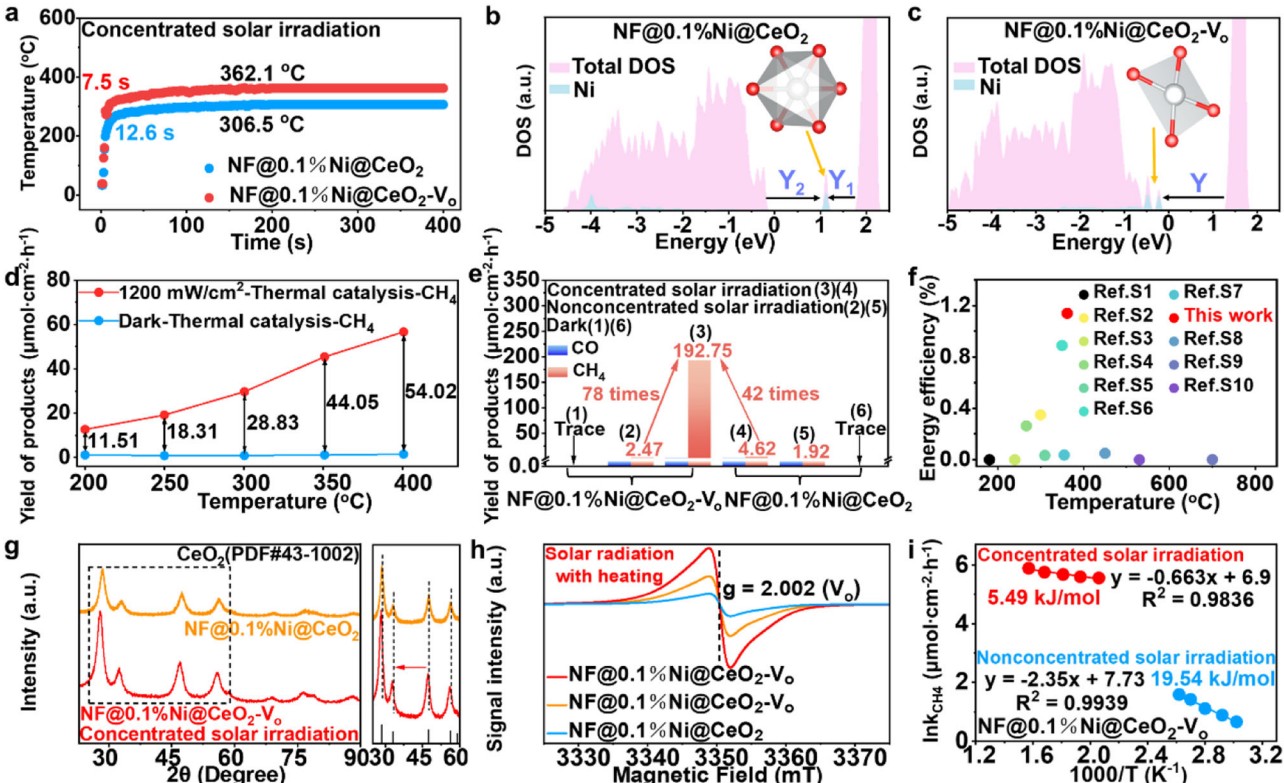

**Fig. 2 | Impact of Ni coordination environments on photo-thermal catalytic performance. a** Surface temperature of NF@0.1%Ni@CeO$_2$ and NF@0.1% Ni@CeO$_2$-V$_o$ catalysts under concentrated solar irradiation (4200 mW/cm$^2$). **b**, **c** Density of states for NF@0.1%Ni@CeO$_2$ (**b**) and NF@0.1%Ni@CeO$_2$-V$_o$ (**c**). **d** Thermocatalytic yields of CH$_4$ from the NF@0.1%Ni@CeO$_2$-V$_o$ catalyst in the dark and with the addition of 1200 mW/cm$^2$ solar light. **e** Photo-thermal catalytic CH$_4$ yields from NF@0.1%Ni@CeO$_2$ and NF@0.1%Ni@CeO$_2$-V$_o$ catalysts under different conditions (non-concentrated solar irradiation represents 420 mW/cm$^2$, and concentrated solar irradiation represents 4200 mW/cm$^2$). **f** Comparison of energy efficiency of a series of literature-reported catalysts under similar conditions. **g**–**i** In-situ XRD pattern (**g**), in-situ EPR pattern (**h**), and apparent activation energy of NF@0.1%Ni@CeO$_2$-V$_o$ (**i**) under concentrated solar irradiation and non-concentrated solar irradiation.

densities can excite electrons in the catalyst, causing the lattice atoms to leap to the defect sites. These leaps can lead to the formation of defects, which can alter the distances between atoms in the crystal, resulting in a change in the lattice constant and shift of the diffraction peak (Fig. 2g).

In situ, electron paramagnetic resonance (EPR) also provided compelling evidence for the increase in V$_o$ in the catalysts under concentrated solar irradiation conditions (Fig. 2h), which enhances the recombination between electrons and holes. Photogenerated carriers can be captured by Ni single atoms and release high-frequency phonons via recombination. Meanwhile, Ni single atoms and V$_o$ form atomic-scale active sites. These active sites, combined with the hot-spot effect induced by carriers' recombination, promote the CO$_2$ reduction reaction. The surface temperature elevation with the introduction of Ni single atoms and V$_o$ is shown in Supplementary Fig. 35a. Besides, V$_o$ indeed possesses the capability to capture photogenerated charges (Supplementary Fig. 35b, c). On the other hand, the in-situ XPS spectrum demonstrates the valence state change of Ni was undetectable due to its low content of 0.1% (Detailed description in Supplementary Fig. 36a). Conversely, the content of Ce$^{3+}$ increased with light irradiation. This result also suggests the increased V$_o$ content under concentrated solar irradiation (Supplementary Fig. 36b).

Additionally, we determined the apparent activation energy for the CO$_2$ reduction to CH$_4$ under concentrated and non-concentrated solar irradiation conditions, respectively. The relation between ln(r$_{CH4}$) and 1/T is plotted in Fig. 2i and Supplementary Table 9, where the measured points are fitted as straight lines using the Arrhenius equation[27,28]:

$$\ln k = \ln A - E_a/RT \qquad (1)$$

where k, R, T, E$_a$ and A denote the rate constant, molar gas constant, thermodynamic temperature, apparent activation energy, and frequency factor, respectively. A smaller slope represents a lower E$_a$. The results suggest that the CH$_4$ production increases owing to a decrease in E$_a$ from 19.54 kJ/mol under non-concentrated solar irradiation to 5.49 kJ/mol under concentrated solar irradiation conditions. The added heat increases the probability of reactant molecules overcoming the activation energy barrier during CO$_2$ reduction reactions. The equation for the Boltzmann distribution is as follows[27]:

$$E_t = \frac{3}{2}k_B T \qquad (2)$$

where E$_t$, k$_B$ and T denote the molecular average kinetic energy, Boltzmann constant and temperature, respectively. E$_t$ increased according to this equation with increasing temperature. This was confirmed by determining the CH$_4$ yield at different temperatures under a constant concentrated solar irradiation intensity (Supplementary Figs. 37, 38; Supplementary Tables 10, 11). The CH$_4$ yield increased with increasing temperature, demonstrating the auxiliary role of heat in the catalysis. Photo-electrons generated by concentrated solar irradiation can further reduce the E$_a$ of the product, and heat can increase the E$_t$ (Supplementary Fig. 39a–d).

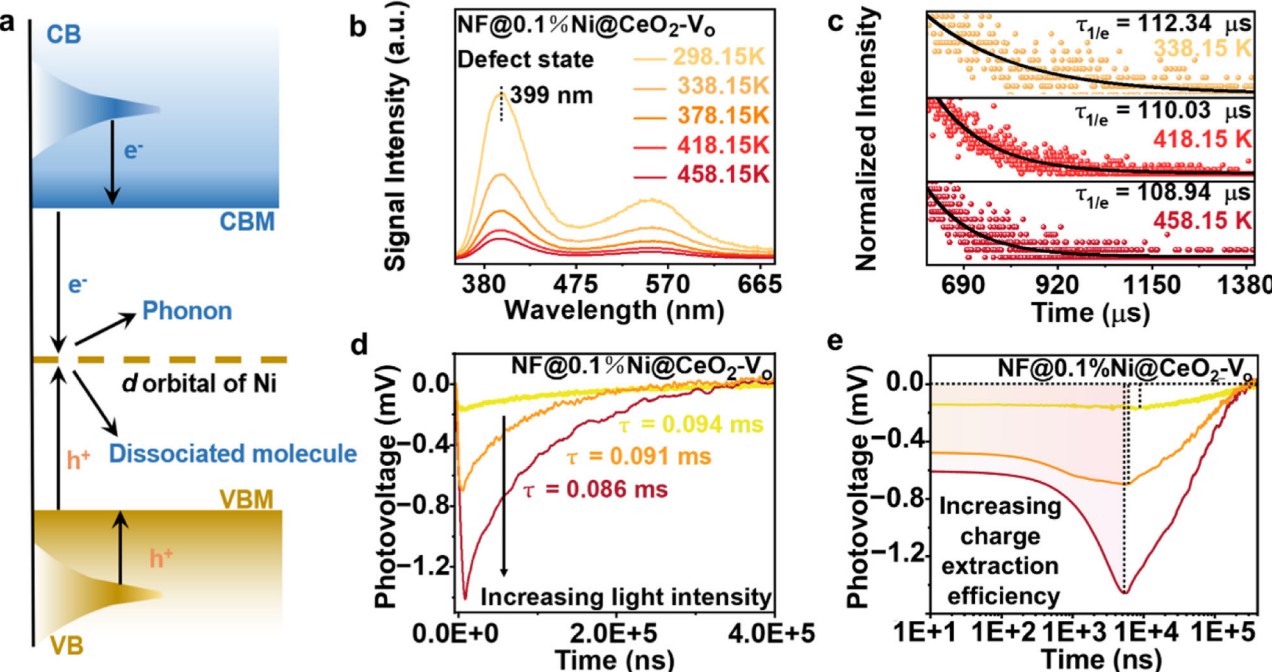

**Fig. 3 | Light and thermal effects on the catalytic mechanism. a** Schematic diagram of two paths of photon energy conversion. **b**–**e** Steady-state PL spectra at different temperatures. **b** TRPL at different temperatures **c** transient surface photovoltage at different irradiation intensities **d** transient surface photovoltage at a different irradiation intensity, obtained by taking the logarithm of the abscissa of 3d (**e**) for NF@0.1%Ni@CeO$_2$-V$_o$.

## Mechanism of light and thermal effects over the Ni single atom

The photo-thermal catalytic $CO_2$ reduction unavoidably involves both thermal and optical effects on the catalyst. Therefore, we comprehensively examined the light and thermal effects on the catalytic mechanism separately. When photo-excitation occurs from the valence band (VB) to the conduction band (CB), a hole relaxes to the VBM and an electron relaxes to the CBM (Fig. 3a). In the presence of a mid-gap state, the electron and hole may directly combine through an intermediate impurity state that captures both charge carriers, forming a thermally dissociated molecule while the photo-generated electron–hole pair eventually become phonons (i.e., heat). However, if only a hole or an electron is captured by the intermediate impurity state, it is rapidly transferred to the adsorbed molecule through the $d$ orbital.

Subsequently, we investigated in detail the migration of carriers induced by the thermal and light effects in NF@0.1%Ni@CeO$_2$-V$_o$. Variable-temperature PL showed that the fluorescence intensity progressively decreases with increasing temperature (Fig. 3b), suggesting that the thermal movement of the carriers increases, thus increasing carrier mobility and separation. Moreover, the fluorescence lifetime gradually decreases with increasing temperature (Fig. 3c) because heat enhances the carrier recombination rate. However, because the migration rate is greater than the complexation rate, heat is favourable for the reaction. OCP tests at variable temperatures also validate this mechanism (Supplementary Fig. 40). Meanwhile, the photo-response of NF@0.1%Ni@CeO$_2$-V$_o$ was the most substantial under concentrated solar irradiation conditions (Supplementary Figs. 41, 42), suggesting that the high photoelectron flux density drives single-atom Ni and V$_o$ to further enhance the conductivity and carrier separation efficiency. Furthermore, the photo-generated electrons under concentrated solar irradiation not only increase in quantity but also exhibit enhanced quality, which can be attributed to the increase in the Fermi energy level in NF@0.1%Ni@CeO$_2$-V$_o$ (Supplementary Fig. 43). The elevated Fermi energy level indicates an increased carrier density in the CB, which is conducive to the photo-thermal reduction of $CO_2$.

Furthermore, we evaluated the TPV curves for different irradiation intensities (Fig. 3d). The electron lifetime decreases with increasing irradiation intensity, which can be attributed to the Auger loss under the high density of excited electrons[27]. For NF@0.1% Ni@CeO$_2$-V$_o$, numerous electrons are generated after being excited by concentrated solar irradiation, some of which migrate and participate in the reduction reaction. Others undergo Auger recombination, releasing the rest of the energy, which is transferred to a third charge carrier that thermalises to the edge of the CB by emitting phonon vibrations, raising the catalyst and facilitating the photo-thermal conversion. We applied logarithmic processing to the tails of the TPV curves; the resulting combined shaded area at different irradiation intensities suggests that the charge extraction efficiency improved with increasing irradiation intensity (Fig. 3e).

## Charge carrier dynamics

To investigate the enhancement of the electron transfer in single-atom Ni under light irradiation, we prepared a NF@CeO$_2$-V$_o$ catalyst without Ni. Its physical phases and morphology are presented in Supplementary Fig. 44. Regardless of the presence of V$_o$, the surface temperature of the catalyst without Ni was lower than that of the Ni-based catalyst (Supplementary Fig. 45). This can be attributed to the localised property of the $d$ orbital of Ni, which enables the capture of photo-generated charge carriers and their efficient conversion into phonons. Furthermore, this finding provides additional evidence for the local hot-spot effect of single-atom Ni. Then, we evaluated the catalytic performance of NF@CeO$_2$-V$_o$ under dark, non-concentrated and concentrated light conditions (Fig. 4a), finding that under concentrated solar irradiation conditions, NF@CeO$_2$-V$_o$ afforded a $CH_4$ yield of 3.88 μmol/cm$^2$/h, which was 50 times lower than that of NF@0.1% Ni@CeO$_2$-V$_o$. This remarkable enhancement suggests that single-atom Ni promotes the separation and transfer of photo-generated carriers. The energy level positions of NF@CeO$_2$-V$_o$, NF@0.1%Ni@CeO$_2$ and NF@0.1%Ni@CeO$_2$-V$_o$ were calculated via UV–vis DRS and valence-band X-ray photoelectron spectroscopy (XPS) using the Kubelka–Munk function (Supplementary Figs. 46, 47), revealing the

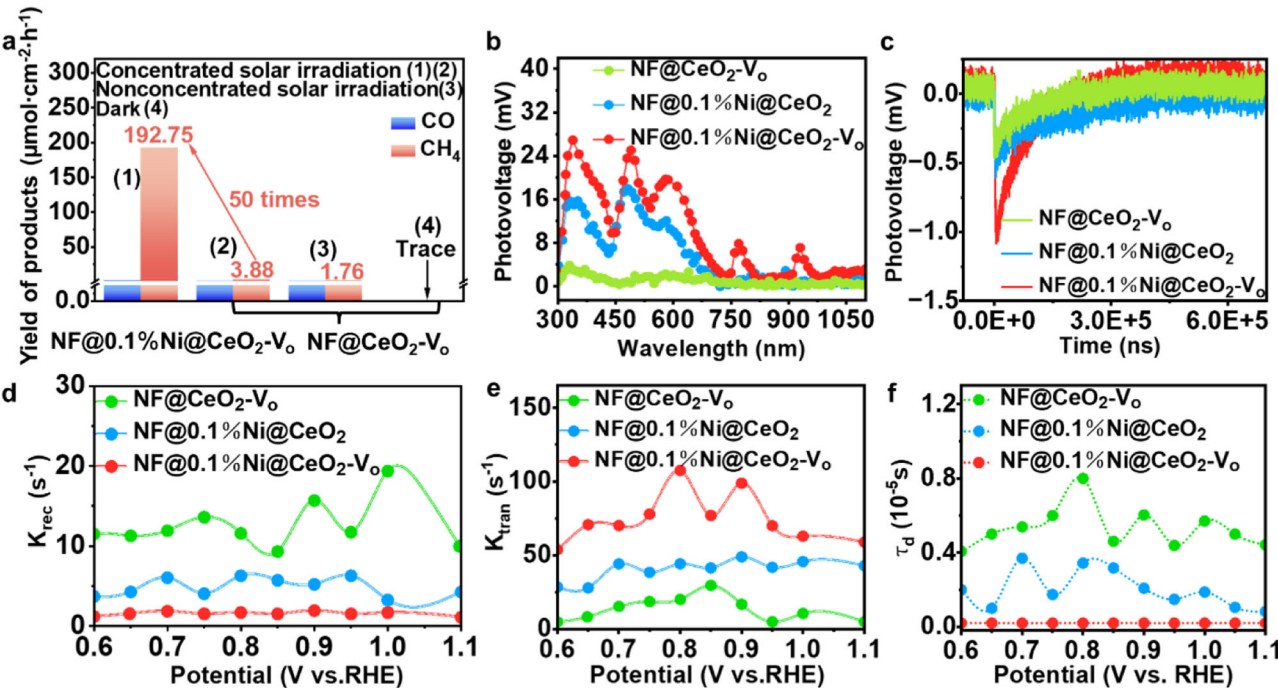

**Fig. 4 | Catalytic performance of the Ni-based catalysts and charge carrier dynamics. a** Photo-thermal catalytic $CO/CH_4$ yields from NF@0.1%Ni@CeO$_2$-V$_o$ and NF@CeO$_2$-V$_o$ catalysts under different conditions. **b–f** Surface photovoltage (**b**), transient surface photovoltage (**c**), plot of the rate constant of charge recombination vs. potential (**d**), plot of the rate constant charge transfer rate vs. potential (**e**), and plot of the average lifetime of photoinduced electrons ($\tau_d$) vs. potential (**f**) for NF@CeO$_2$-V$_o$, NF@0.1%Ni@CeO$_2$, and NF@0.1%Ni@CeO$_2$-V$_o$.

consistency of the band gaps with the potential for the reduction of $CO_2$ to $CH_4$.

The results suggest the occurrence of considerable synergistic effects between V$_o$ and single-atom Ni. The $CO_2$ temperature-programmed desorption spectra shown in Supplementary Fig. 48 confirm that V$_o$ and Ni can provide surface basic sites to facilitate the $CO_2$ adsorption and decomposition. To investigate the recombination of photo-generated carriers, steady-state photoluminescence (PL) tests were conducted using an excitation wavelength of 255 nm (Supplementary Fig. 49a). Both NF@CeO$_2$-V$_o$ and NF@0.1%Ni@CeO$_2$-V$_o$ produced a sharp convex peak at approximately 400 nm, which can be attributed to defective-state luminescence[29]. Moreover, the photo-generated carriers in NF@0.1%Ni@CeO$_2$-V$_o$ are highly resistant to recombination, resulting in more electrons available for the $CH_4$ production reaction. Supplementary Fig. 49b demonstrates that among all samples, NF@0.1%Ni@CeO$_2$-V$_o$ showed the longest emission lifetime (114.42 μs), as measured using time-resolved PL spectroscopy. The values of $\tau_1$, $\tau_2$ and average emission lifetime ($\tau$) were obtained through data fitting (Supplementary Table 13). The synergistic effect of Ni and V$_o$ was found to enhance the electron transport efficiency at the catalyst surface, prolonging the excited state of the fluorescent molecules and thereby substantially reducing the carrier recombination. The synergistic effect of Ni and V$_o$ was found to enhance the electron transport efficiency at the catalyst surface, prolonging the excited state of the fluorescent molecules and thereby substantially reducing the carrier recombination. This phenomenon arises due to the presence of a protective charge, preventing the emission of fluorescence when carriers are captured by the trap state. Therefore, while the overall fluorescence intensity decreases, carrier recombination resulting in fluorescence forms a longer temporal tail. Additionally, this protective charge increases the carrier lifetime by localizing around the trap state and reducing spatial overlap with other charge carriers. The above analysis explains why V$_o$ acting as a trap state leads to weaker fluorescence intensity yet longer carrier lifetime.

Subsequently, to obtain information about the charge separation direction, we investigated the photo-electrical properties of the catalysts by measuring the surface photo-voltage (SPV) signal generated by the spatial separation of photo-generated charge carriers. NF@0.1% Ni@CeO$_2$-V$_o$ exhibited a strong response signal in the range of 300–1000 nm compared with other catalysts (Fig. 4b) because Ni and V$_o$ can accept photo-generated charge carriers from CeO$_2$ through interface charge transfer, improving the separation efficiency. Transient surface photo-voltage (TPV) spectroscopy measures the SPV signal as a function of the photon energy and relaxation time[30]. For the $CO_2$ reduction reaction, the photo-generated electrons must migrate towards the outer surface, generating SPV signal peaks in the downward direction. The TPV curves of the catalysts (Fig. 4c) revealed that NF@0.1%Ni@CeO$_2$-V$_o$ exhibits the largest photo-voltage and shifts towards a long-time scale (Supplementary Table 14). Furthermore, we logarithmically processed the tails of the TPV curves and observed a general rightward shift in the curves (Supplementary Fig. 50), further confirming that the co-existence of Ni and V$_o$ alters the charge separation channel and enhances the separation of photo-induced electron and hole pairs. This was also supported by linear sweep voltammetry (LSV) and electrochemical impedance spectroscopy (EIS) measurements (Supplementary Fig. 49c, d).

Investigating the behaviour of photo-induced carriers on the Helmholtz layer is also essential to reveal the charge transfer and complexation dynamics in NF@0.1%Ni@CeO$_2$-V$_o$. Therefore, we used intensity-modulated photocurrent spectroscopy (IMPS) to evaluate the pseudo-first-order rate constants for surface complexation ($k_{rec}$) and hole transfer ($k_{tran}$)[31]. Supplementary Fig. 51 displays the typical IMPS responses of different catalysts in the complex plane. The $k_{rec}$ and $k_{tran}$ values were calculated using the IMPS spectra obtained at different applied bias voltages. NF@0.1%Ni@CeO$_2$-V$_o$ showed lower $k_{rec}$ and higher $k_{tran}$ values than NF@CeO$_2$-V$_o$ and NF@0.1%Ni@CeO$_2$, indicating that the Ni-and-V$_o$–containing catalyst exhibited improved charge separation/transfer kinetics (Fig. 4d, e). Specifically, the lack of

a substantial increase in the $k_{tran}$ values of NF@CeO$_2$-V$_o$ and NF@0.1% Ni@CeO$_2$ suggests that single-atom Ni and V$_o$ are the main contributors to the charge transfer process. The average photo-induced electron transfer time ($\tau_d$) estimated using the lowest imaginary frequency was shorter for NF@0.1%Ni@CeO$_2$-V$_o$ (Fig. 4f), demonstrating its superior charge transfer kinetics. Furthermore, we conducted IMPS measurements on NF@0.1%Ni@CeO$_2$-V$_o$ under light sources of different wavelengths (Supplementary Fig. 52a). The $\tau_d$ value was minimised after UV–visible irradiation (Supplementary Fig. 52b), suggesting that photo-thermal coupling conditions promoted the charge separation and transfer on NF@0.1%Ni@CeO$_2$-V$_o$.

## CO$_2$ reduction mechanism under concentrated solar irradiation

In the photo-thermally coupled catalytic reaction of CO$_2$ and H$_2$O, the activation of H$_2$O is the primary rate-limiting step (Supplementary Fig. 53a). According to the thermodynamic state, H$_2$O decomposition requires a high energy barrier to proceed with the CO$_2$ reduction reaction (Supplementary Fig. 53b, c). The Gibbs free energy for H$_2$ and O$_2$ production from H$_2$O decomposition is 273 KJ mol$^{-1}$, indicating that the reaction cannot proceed spontaneously and requires additional energy to occur. Therefore, to identify the active site responsible for H$_2$O activation, we employed ab initio molecular dynamics (AIMD) simulations to investigate the desorption of CO$_2$ on a single-atom Ni active site (Fig. 5a). The results showed that the C in CO$_2$ cannot coordinate to the Ni sites of the catalyst surface because they are positively charged. Therefore, CO$_2$ is not stably adsorbed to the surface by bonding single-atom Ni. The CO$_2$ adsorption energy confirms this view (Detailed description in Supplementary Figs. 54, 55). Furthermore, when CO$_2$ molecules adsorb to the Ni active site, desorption occurs within approximately 100 fs, indicating that Ni is not an effective active site for CO$_2$ adsorption.

In-situ DRIFTS spectroscopy was used to identify the critical reaction intermediates generated during the catalytic reaction (Fig. 5b). The remarkably strong m-CO$_3^{2-}$ peaks indicate that CO$_2$ is adsorbed on the lattice O of CeO$_2$ instead of on Ni. The production of –CH$_2$O, –CH$_3$O and –CH$_3$ intermediates confirms the migration of multiple photo-generated electrons and, notably, the production of CH$_4$[32,33]. The adsorption bands at 3725, 3677, 3624 and 3594 cm$^{-1}$ are attributed to physical adsorption via hydrogen-bound OH groups and H$_2$O stretching vibrations (Supplementary Fig. 56)[34]. Upon adsorption of a H$_2$O molecule to single-atom Ni, the structure undergoes relaxation, resulting in a deformed tetrahedral structure. Hydrogen bonding occurs between the two H atoms of the H$_2$O molecule and the two lattice O atoms originally coordinated to Ni. Near the VBM, the molecular orbital of the H$_2$O molecule hybridises with the $d$ orbital of Ni, forming a bridge to transfer the photo-generated charge carriers. This hybridised state is an occupied state; the photo-generated holes on the VBM are subsequently transferred to the H$_2$O molecule through Ni (Fig. 5c).

If only thermal dissociation (600 K) is considered after the adsorption of H$_2$O molecules on the catalyst surface, the H$_2$O on the CeO$_2$ surface would exist in a dissociated state and the released O would fill the V$_o$ around single-atom Ni directly after the dissociation of H$_2$O molecules. However, H would still be trapped near the lattice O and not transfer to CO$_2$, resulting in low reactivity (Fig. 5d). In the absence of light excitation and at a temperature of only 10 K, H$_2$O molecules are stably adsorbed (Fig. 5e). However, H$_2$O molecules are dissociated upon applying an intense laser field. Owing to the presence of captured photo-generated holes on the H$_2$O molecules, the released O species are rapidly desorbed from the Ni sites (Fig. 5f). H species are then transferred to CO$_2$ and captured by an O atom, leading to CO$_2$ reduction and completing the methanation process. The adsorption energies of H$_2$O on V$_o$ and Ni single atoms are −0.67 eV and −1.47 eV, respectively (Supplementary Fig. 57). It is worth noting that on CeO$_2$-

V$_o$, H$_2$O is adsorbed by the unsaturated Ce atoms instead of V$_o$ sites. Meanwhile, on Ni@CeO$_2$, H$_2$O is adsorbed by the Ni single atoms instead of Ce atoms (Supplementary Fig. 58). Additionally, H$_2$O-TPD demonstrates that the incorporation of Ni single atoms and V$_o$ can significantly enhance the dissociation capacity of H$_2$O (Supplementary Fig. 59). Any remaining O species may form O$_2$ molecules with other O species or fill the V$_o$, leading to Ni deactivation, which could shift from the planar quadrilateral coordination structure to an octahedral one. To confirm the mechanism, we assessed the concentration of oxygen vacancies before and after the reaction. The findings revealed an increase in the concentration of oxygen vacancies after the reaction, thereby implying that the oxygen vacancies were not occupied by O$_2$ (Supplementary Fig. 60). Moreover, the O$_2$ yield reached 762.5 μmol/ cm$^2$ of concentrated solar irradiation (Supplementary Fig. 61). A slight decrease in activity was observed after six catalytic cycles. The microstructural changes upon H$_2$O dissociation during the thermally induced and laser-induced reactions are illustrated in Supplementary Fig. 62.

Additionally, we conducted time-dependent DFT (TDDFT) simulations to observe changes in the absence of a catalyst. The application of a laser field resulted in no dissociation of small molecules, indicating that the dissociation of H$_2$O molecules is due to the electron–hole excitation in the catalyst and not the direct action of the laser field (Supplementary Figs. 63, 64). We also performed three cycles of photo-thermal coupled CO$_2$ reduction experiments. Although a considerable decrease in the CH$_4$ yield was observed after the first cycle, a substantial increase occurred in the second and third cycles upon the introduction of hydrogen (Fig. 5g). This phenomenon may be attributed to the regenerative function of V$_o$ under concentrated light conditions, providing a continuous supply of active sites for CO$_2$ reduction. This hypothesis was verified by subjecting NF@CeO$_2$-V$_o$ and NF@0.1%Ni@CeO$_2$-V$_o$ to valence-band XPS under different conditions (Supplementary Figs. 65, 66 and Supplementary Table 15), as discussed in detail in Supplementary Fig. 67.

A plausible mechanism for the photo-thermal reduction of CO$_2$ catalysed by NF@0.1%Ni@CeO$_2$-V$_o$ under concentrated solar irradiation conditions is depicted in Fig. 5h. Initially, CO$_2$ is adsorbed to the surface lattice O of CeO$_2$, while H$_2$O molecules are adsorbed on single-atom Ni. The thermally assisted photo-dissociation of H$_2$O by Ni generates H, which converts CO$_2$ to CH$_4$ and O to O$_2$, with the concomitant release of the products from the catalyst surface. This can be illustrated in detail using a Feynman diagram, where W represents the photons that excite the CB and VB, releasing electrons and holes, respectively. Some holes are responsible for the photo-dissociation of H$_2$O molecules, while other holes combine with electrons through the intermediate impurity state of Ni, forming phonon Y, which facilitates H$_2$O dissociation through heating.

## Discussion

To conclude, we synthesized a high-performance NF@0.1% Ni@CeO$_2$-V$_o$ catalyst with single-atom Ni anchored to CeO$_2$ porous nanorods containing V$_o$. This catalyst demonstrated CH$_4$ production with 192.75 μmol/cm$^2$/h yield, ~100% selectivity, and 1.14% efficiency in direct STC energy conversion under concentrated solar irradiation. This irradiation triggers V$_o$ regeneration on the catalyst surface, ensuring a continuous supply of active sites for CO$_2$ reduction. The hybridized molecular orbitals of H$_2$O and single-atom Ni, along with structural changes induced by V$_o$, enhance charge carrier trapping and activation of reactant molecules, while thermal- and photo-dissociation of H$_2$O on the catalyst surface was confirmed through computational and spectroscopic studies. This research provides insights into the development of highly efficient systems for direct STC energy conversion via the CO$_2$/H$_2$O reaction and the design of catalysts for CO$_2$ conversion under concentrated solar irradiation.

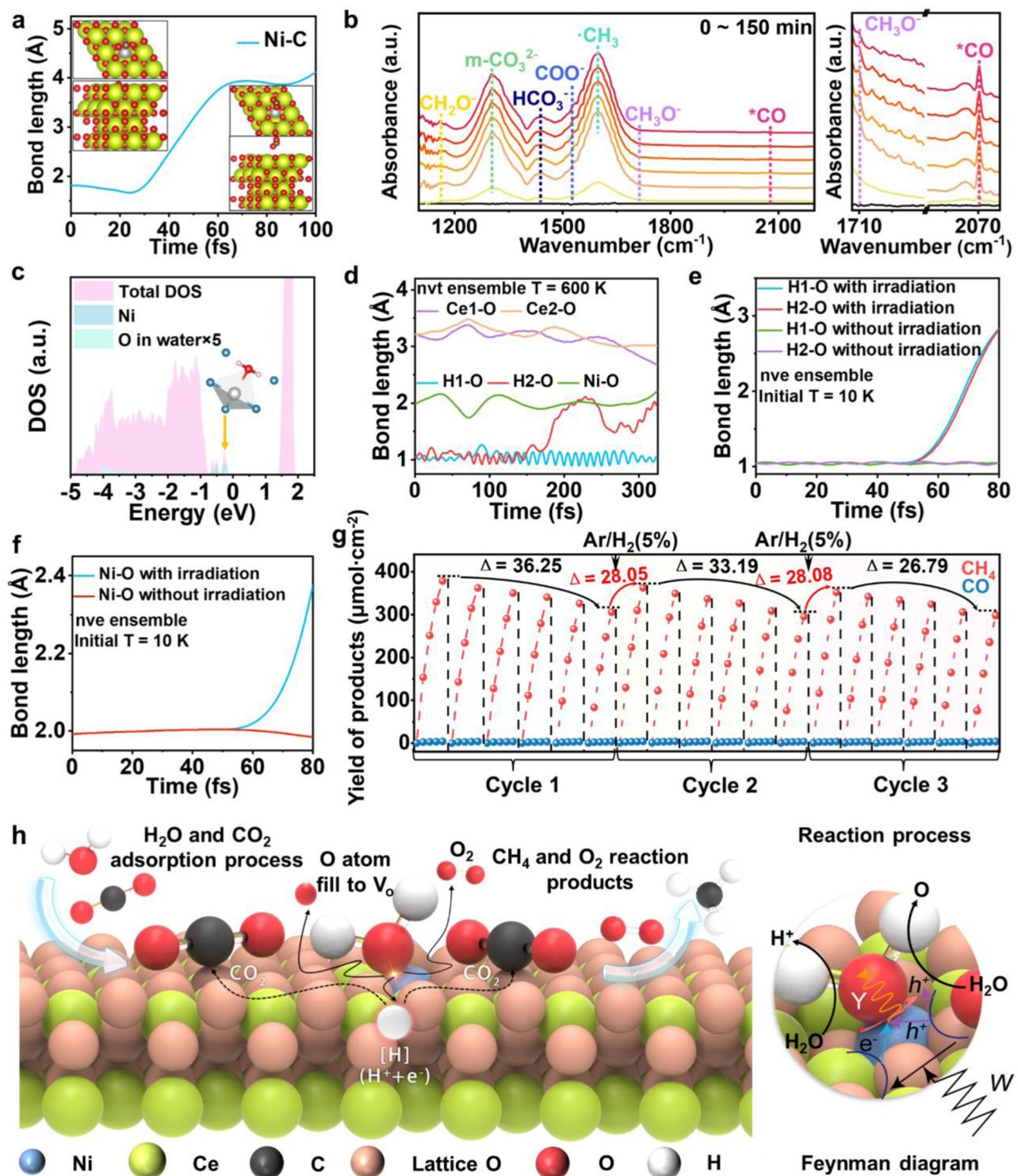

**Fig. 5 | Investigation into CO₂ reduction mechanism by NF@0.1%Ni@CeO₂-Vₒ under concentration solar irradiation. a** AIMD simulation of CO₂ desorbing from Ni active site. **b** In-situ DRIFTS spectra of NF@0.1%Ni@CeO₂-Vₒ under CO₂/H₂O conditions and 1000-2200 cm⁻¹ spectra. **c** Density of states for NF@0.1%Ni@CeO₂-Vₒ with H₂O adsorption. **d** AIMD simulation of heat-induced H₂O dissociation. **e, f** TDDFT simulation of laser-induced H₂O dissociation. **g** Cycling stability profile after H₂ exposure (each cycle is 12 h). **h** Schematic illustration of the mechanism of NF@0.1%Ni@CeO₂-Vₒ for CO₂ reduction upon concentrated solar irradiation.

## Methods

### Materials and reagents

All chemicals utilized in this study, including cerium nitrate hexahydrate (Ce(NO₃)₃·6H₂O, analytical reagent grade (AR), 99.95%), sodium hydroxide (NaOH, AR, 95%), nickel nitrate hexahydrate (Ni(NO₃)₂·6H₂O, AR, 98%), sodium borohydride (NaBH₄, 98%), sodium tungstate dihydrate (Na₂WO₄·2H₂O, 99.5%), hydrazine monohydrate (N₂H₄·H₂O, >98%), hydrochloric acid (HCl, 37wt.% in H₂O, 99.999%), zirconyl chloride hydrate (ZrOCl₂·nH₂O, 99.99%), hydrofluoric acid (HF, 48wt.% in H₂O, 99.99%), tetraisobutyl titanate (C₁₆H₃₆O₄Ti, 99.5%) were purchased from Aladdin and directly employed without any additional purification steps. The Ni foam substrate was procured from

Kun Shan Electronic Material Co., Ltd. (China). A mixed gas mixture composed of carbon dioxide and nitrogen ($CO_2$: $N_2$ = 1:99, v:v) was obtained from the Nanjing Shangyuan industrial gas plant, while all aqueous solutions used during the experimental procedures were prepared using ultrapure water.

## Synthesis

**Preparation of the monolithic NF@NP-CeO$_2$.** In this study, we synthesized $CeO_2$ non-porous nanorods using an improved method[35], which involved the use of nickel foam as a loading substrate for monolithic catalysts. The nickel foam used had a thickness of 2 mm and dimensions of 1 cm × 1 cm. Before use, it was sonicated in hydrochloric acid solution (10 mL, 0.1 M) for 30 min, followed by sonication in deionized water (10 mL) for another 30 min to eliminate any organic carbon impurities that may have been present on the surface of the nickel foam. It was then dried in a vacuum oven at 70 °C for 1 h. To prepare the $CeO_2$ precursor solution, $Ce(NO_3)_3 \cdot 6H_2O$ (1.375 g) was dissolved in deionized water (10 mL) and agitated at 600 r/min for 30 min. Separately, NaOH (9.5 g) was dissolved in deionized water (70 mL) and agitated at 600 r/min for 30 min. The $Ce(NO_3)_3 \cdot 6H_2O$ solution was slowly added drop by drop into the NaOH solution while being vigorously stirred. The treated nickel foam was then added to the solution, and the mixture was agitated at 600 r/min for 2 h. The nickel foam mixture solution containing the precipitate was transferred to a Teflon-lined reactor, sealed, and heated to 100 °C for 24 h. The resulting non-porous cerium oxide grown on nickel foam and non-porous cerium oxide powder were each washed three times each with ethanol and water, respectively. They were then dried in a vacuum oven at 70 °C for 10 h. The monolithic catalysts obtained were named NF@NP-CeO$_2$.

**Preparation of the monolithic NF@0.1%Ni@CeO$_2$-V$_o$, NF@0.1% Ni@CeO$_2$ and NF@CeO$_2$-V$_o$.** To prepare the NF@0.1%Ni@CeO$_2$-V$_o$ catalyst, nickel nitrate hexahydrate (9.91 mg) and a certain amount of sodium borohydride were dissolved in 10 mL of deionized water and sonicated for 1 h. Next, non-porous cerium oxide powder (2 g) and a certain amount of monolithic catalyst NF@NP-CeO$_2$ were added to the mixed solution and sonicated for 2 h. The resulting mixture solution was transferred to a Teflon-lined reactor, sealed, and heated at 160 °C for 14 h. Finally, the porous cerium oxide nanorods grown on nickel foam were washed three times each with ethanol and water, respectively, and dried in a vacuum oven at 70 °C for 8 h. The sample obtained was named NF@0.1%Ni@CeO$_2$-V$_o$. The preparation recipe for NF@0.1%Ni@CeO$_2$ catalyst is similar to that for NF@0.1%Ni@CeO$_2$-V$_o$ catalyst, except that hydrothermal conditions (110 °C, 14 h) were used. The NF@CeO$_2$-V$_o$ catalyst was prepared similarly to the NF@0.1% Ni@CeO$_2$-V$_o$ catalyst, but without adding $Ni(NO_3)_2 \cdot 6H_2O$. Both NF@0.1%Ni@CeO$_2$-V$_o$ and NF@0.1%Ni@CeO$_2$ had 0.1% Ni content measured by inductively coupled plasma mass spectrometry (ICP-MS) (Supplementary Table 15).

**Preparations of WO$_3$.** 1 mmol $Na_2WO_4 \cdot 2H_2O$ along with nickel foam (1 cm × 1 cm × 2 mm) were dissolved in 20 mL $H_2O$ and the solution was then stirred for 1 h. Next, 0.5 mmol $N_2H_4 \cdot H_2O$ was added to the solution and stirred for 1 h. The pH of the mixture solution was then adjusted to 3 using concentrated hydrochloric acid and the solution was stirred for an additional 0.5 h. Afterwards, the mixture solution was heated at 180 °C for 12 h. Finally, the mixture solution was centrifuged, washed, and dried, which was labelled as WO$_3$.

**Preparations of ZrO$_2$.** We dissolved 8 g of $ZrOCl_2 \cdot nH_2O$ and nickel foam (1 cm × 1 cm × 2 mm) in 70 mL of deionized water and stirred the solution for 30 min at 600 r/min. Next, we transferred the solution to a Teflon autoclave, sealed it, and heated it to 150 °C for 6 h. After cooling the samples to room temperature, we separated the product by

centrifugation, washed it three times with deionized water, and dried it in a vacuum oven at 70 °C for 10 h. This sample was labeled as $ZrO_2$.

**Preparations of TiO$_2$.** A mixture of 1.5 mL of HF solution and 15 mL of $C_{16}H_{36}O_4Ti$ solution was stirred for 1 h. Subsequently, a solvothermal reaction was carried out at 200 °C for 24 h in a 25 mL Teflon autoclave. After natural cooling to room temperature, the precipitants were collected and fully washed with distilled deionized $H_2O$, then dried in a vacuum oven at 70 °C for 10 h. Then, the powder and nickel foam (1 cm × 1 cm × 2 mm) were stirred in deionized water for 1 h and then aged for 2 h. Finally, the $TiO_2$ nanosheets were annealed for 3 h in air atmosphere at 450 °C with the temperature elevation rate of 5 °C/min.

## Characterizations

High-resolution transmission electron microscopy (HRTEM) images and the corresponding energy-dispersive spectroscopy mapping analyses were obtained using a Talos F200X system. HAADF-STEM images and the images with atomic resolution of samples were performed using a Titan Cubed Themis 60-300 operated at 300 kV, equipped with a probe spherical aberration corrector. X-ray diffraction (Bruker D8-Discover diffractometer, Cu-Kα radiation ($\lambda$ = 0.1542 nm)) and in situ, X-ray diffraction (Nippon Rigaku Ultimate IV) was used to investigate the crystal phase structure, scanned in the range of 2θ = 10–90 with a scanning rate of 5° min$^{-1}$, specific test temperature range: 25–400 °C in an $N_2$ atmosphere. X-ray photoelectron spectroscopy (XPS) spectra were collected using an ESCALab MKII X-ray photoelectron spectrometer, and all binding energies were calibrated using the contaminant carbon (C1s = 284.6 eV) as a reference. X-ray absorption spectra (XAS) including X-ray absorption near-edge structure (XANES) at Ni K-edge and Ce L$_3$-edge of the samples and extended X-ray absorption fine-structure (EXAFS) at Ni K-edge and Ce L$_3$-edge of the samples were measured at the beamline 14 W of Shanghai Synchrotron Radiation Facility (SSRF) in China. The output beam was selected by Si (311) monochromator, and the energy was calibrated by Ni foil, NiO, $Ni_2O_3$, and $CeO_2$. The data were collected at room temperature under transmission mode. Athena software package was employed to process the XAS data. The specific surface area was calculated using a Brunauer-Emmett-Teller (BET) method. Electron paramagnetic resonance (EPR, Bruker EMXPLUS-6/1) spectroscopy and in situ, electron paramagnetic resonance (EPR, Bruker EMXPLUS-6/1) were used to detect the oxygen-vacancy signal. UV–Vis diffuse reflectance spectra (UV-2600, Shimadzu) were used to study the optical properties of the different catalysts at 300–2500 nm. Catalyst surface temperature and environment temperatures were determined by thermocouple and infrared thermal imager (FLIR). Simultaneous thermal analysis TG-DSC (TA) tests were used to investigate the thermal stability of the catalysts in an $N_2$ atmosphere at a ramp-up rate of 10 °C/min from room temperature to 600 °C. The $CO_2$ temperature-programmed desorption measurement ($CO_2$-TPD, AUTO CHEM 2920) was used to confirm the active adsorption sites of the catalysts. In-situ diffuse reflectance infrared Fourier transform spectroscopy (DRIFTS, Bruker TENSOR II) measurements were also performed to investigate the intermediate during the $CO_2$ reduction reaction. Considering the limited experimental conditions, the catalysts were carefully extracted from the nickel foam substrate and tested by XAS, XPS, HRTEM, BET, TG-DSC, surface photovoltage (SPV), and transient surface photovoltage (TPV), PL, TRPL, EPR, in-situ DRIFTS, and $CO_2$-TPD.

## Details of the photocatalytic CO$_2$ reduction

$CO_2$ photocatalytic reduction experiments were carried out in a 100 mL laterally irradiated heat-resistant closed concentrated solar reactor. Light irradiation was provided by a PLS-SXE300/300UV Xe lamp (Beijing China Education Au lamp) with a standard AM 1.5 G filter and simulated sunlight using an AM 1.5 G filter with an output optical density of approximately 4200 mW/cm². Two Fresnel lenses (diameter:

30 mm) were used to focus the sunlight. First, the prepared monolithic catalysts were fixed with high-temperature-resistant PTFE clips. Then, water (1 mL) was added as the reactant. In addition, the concentrated solar reactor was purged with $N_2$ for 30 min to ensure a carbon-free atmosphere, and then $CO_2$ (1% $CO_2$ and 99% $N_2$) was passed through for 5 min. Finally, the sealed reactor was left at room temperature, and the gas was analyzed every 30 min using a gas chromatograph (GC-9860-5C). The reaction time for the entire experiment was 2 h.

## Photoelectrochemical measurements

Electrochemical impedance spectroscopy (EIS), instantaneous photocurrent measurements (It), linear scanning voltammetry (LSV), and open circuit voltage (OCP) analyses were performed in a standard three-electrode configuration using an electrochemical workstation (CHI660E). To prepare the working electrode, the catalyst (10 mg) was sonicated and dispersed in a mixture of ethanol (1 mL) and Nafion (15 μL), and subsequently applied to the conductive side of FTO glass (1 × 1 cm$^2$) before being vacuum-dried at 70 °C for 3 h. Platinum foil was used as the counter electrode, Ag/AgCl was used as the reference electrode, and the electrolyte was an aqueous $Na_2SO_4$ solution of 0.1 M. Lighting for the experiments was provided via a high-powered 300 W Xe lamp.

## IMPS measurements

The intensity-modulated photocurrent spectroscopy (IMPS) measurements were conducted in a 1 M KOH solution, utilizing a constant potential meter (Zahner PP211, Zahner, Germany) and a three-electrode setup while varying the bias potentials. For the control IMPS test, high-intensity light-emitting diodes (LED: LSW-2) were utilized as the light source across different wavelengths, including 325 nm in the UV region, 560 nm in the visible region, and 780 nm in the NIR region, with an average intensity of 100 mW cm$^{-2}$. During the IMPS test, the small ac disturbance light was set to 10% of the dc bias light, while the experiments were performed over a frequency range from 10 kHz to 100 MHz under illumination conditions. Furthermore, impedance data were analyzed via a Randles equivalent circuit using Zview software (Scribner Associates).

In a typical IMPS response, the frequency of the maximum imaginary part matches the sum of the rate constants for charge transfer ($k_{tran}$) and recombination ($k_{rec}$) as expressed[31],

$$k_{tran} + k_{rec} = 2\pi f_{max} \qquad (3)$$

Additionally, the ratio of the steady-state photocurrent ($j_{ss}$) to the instantaneous photocurrent ($j_{hole}$) could be used to calculate the hole transfer efficiency ($\eta_{tran}$) at the semiconductor/electrolyte interface. The hole transfer efficiency could also be expressed by the ratio of $k_{rec}$ and $k_{tran}$, assuming that both hole transfer and recombination were pseudo-first-order in the surface hole concentration.

$$\eta_{tran} = \frac{j_{ss}}{j_{hole}} = \frac{k_{tran}}{k_{tran} + k_{rec}} \qquad (4)$$

The average photogenerated electron transfer time ($\tau_d$) could be estimated from the frequency at the minimum imaginary part,

$$\tau_d = \frac{1}{2\pi f_{min}} \qquad (5)$$

## Solar-to-chemical energy conversion efficiency

The solar-to-chemical energy conversion efficiency could be expressed by the input vs. output energy, which can be calculated by the following formula[27]:

$$\eta_{STF} = \frac{qm}{IAt} \times 100\% \left( \frac{J \cdot mol^{-1} \cdot mol}{W \cdot cm^{-2} \cdot cm^2 \cdot s} = \frac{J}{W \cdot s} = 1 \right) \qquad (6)$$

Where q is the combustion heat value of $CH_4$ ($q_{CH_4} = 890.311 0^3 Jmol^{-1}$), m denotes the mole of produced $CH_4$ (mol), I denotes the light intensity (W cm$^{-2}$), A is the irradiation area (cm$^2$), and t denotes the reaction time (s). Thus, the solar-to-chemical energy conversion efficiency for the monolithic NF@0.1%Ni@CeO$_2$-V$_o$ catalyst was calculated to be 1.14%.

For 1 cm$^2$ NF@0.1%Ni@CeO$_2$-V$_o$ catalyst reacting under the concentrated solar (light) reactor for 2 h and producing 385.5 μmol/cm$^2$ $CH_4$:

$$\eta_{STF} = \frac{890.8 \times 10^3 \times 385.5 \times 10^{-6}}{4200 \times 10^{-3} \times 1 \times 7200} \times 100\% = 1.14\% \qquad (7)$$

## SPV measurements

The SPV spectra were obtained based on a lock-in amplifier. The measurement system consisted of a lock-in amplifier (SR830, Stanford Research Systems), a monochromator (190–2000 nm), a light chopper (SR540, Stanford Research Systems), and a sample chamber[36]. Monochromatic light was generated by a 500 W Xe lamp (CHF-XM-500 W, Global Xenon Power) passing through a monochromator (Omni-3007, no. 16047, Zolix).

## Details of PL, TRPL, SPV, and TPV measurements

The excitation wavelength for the PL tests was 255 nm, and the emission lifetime was obtained by fitting the fluorescence attenuation curve according to the following formula:

$$y = y_0 + A_1 e^{-X/t_1} + A_2 e^{-X/t_2} \qquad (8)$$

Moreover, all photovoltage measurements were strictly conducted under the same experimental conditions to ensure the precision and reliability of the experimental outcomes.

## Calculation details

A (3 × 3) supercell of the 111 surfaces of $CeO_2$ was adopted, containing four layers and 108 atoms; the bottom two layers were fixed to maintain the bulk properties. A vacuum layer of 15 Å along the z direction, perpendicular to the surface, was adopted. The Ni atom replaced one Ce atom to simulate the Ni single-atom catalyst.

Ground-state density functional theory (DFT) and ab initio molecular dynamics (AIMD) calculations were conducted with the Vienna Ab initio Simulation Package[37,38], using a projector-augmented wave method and Perdew-Burke-Ernzerhof (PBE) functional[39]. The energy cutoff was set to 500 eV. For Ni and Ce, Ueff = 4.5 and 5.0 eV were employed on the 3$d$ and 4$f$ orbitals, respectively[40–43]. The structures were relaxed until forces were less than 0.001 eV/Å. The total energy convergence criterion for electronic self-consistent iteration was set at $10^{-8}$ eV. A (3 × 3 × 1). Monkhorst-Pack grid k-point mesh was used for sampling the Brillouin zone during relaxation. A (9 × 9 × 1) k-point mesh was used for calculating the electronic structure. The van der Waals interaction was corrected with the DFT-D3 method[44]. During the AIMD simulation, we used the NVT ensemble implemented in VASP[45–48], and only the gamma point was used to sample the Brillouin zone.

Nonadiabatic molecular dynamics (NAMD) with a real-time time-dimensional DFT (TDDFT) method[49,50] was performed to simulate the laser-induced molecular dissociation on the surface of the catalyst. The norm-conserving pseudopotential[51,52] and numerical atomic orbital basis sets[53] with an energy cutoff of 100 Ry (about 1360.5 eV) and PBE

functional were employed for NAMD simulations. Only the gamma point sampled the Brillouin zone. The atoms in catalysts were all fixed during the NAMD simulation to avoid heat-induced molecular dissociation. The initial velocity of adsorbed molecules was initialized using a temperature of 10 K by running an AIMD simulation with the NVT ensemble. Then the NAMD simulation adopted the NVE ensemble and excited the photon-induced chemical reaction by applying an external laser field perpendicular to the surface of the catalyst. The external laser field was shaped by a Gaussian envelope, shown in Supplementary Fig. 63.

$$E(\omega, t) = E_0 \exp\left[-\frac{(t-t_0)^2}{2\tau^2}\right]\cos(\omega t - \omega t_0) \qquad (9)$$

where the $\tau$ was set to 10 fs. The electric field reached the maximum of 1.25 V/Å ($E_0$) at $t_0 = 45$ fs. The frequency ($\omega$) of the laser field was set so that the photon energy $\hbar\omega$ was equal to 4 eV. The real-time TDDFT in the NVE ensemble evolved to 80 fs with a time step of 10 as. The external laser field was applied from $t = 30$ fs to 60 fs.

### Reporting summary
Further information on research design is available in the Nature Portfolio Reporting Summary linked to this article.

## Data availability
All relevant data generated in this study are provided in the Supplementary Information/Source Data file. Source data are provided with this paper.

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

## Acknowledgements

This work was financially supported by the National Natural Science Foundation of China (No. 22078057 and No. 52276212), the Natural Science Foundation of Jiangsu Province (No. BK20221470), the National Natural Science Foundation of China (Key Program of joint fund No. U22A20435), the Fundamental Research Funds for the Central Universities, the Key Research and Development Program in Shaanxi Province of China (No. 2023-YBGY-300), and the Scientific and Technological Innovation Project of Carbon Emission Peak and Carbon Neutrality of Jiangsu Province (No. BK20220001).

## Author contributions

Y.R. prepared the samples, performed all tests related to the experiments, analysed the data and wrote the manuscript. Y.F. and Y.S. participated in the DFT and AIMD calculations. C.Y. and J.H. visualised and analysed the graphs. Z.S., J.Z., K.H. and Y.Z. suggested the experiments. N.L., L.D., and M.L. were responsible for project coordination, project management, and obtaining funding. M.L., W.C., K.H., J.Z., Y.S. and N.L. wrote, reviewed, and edited the manuscript. All authors analysed the data and participated in discussions.

## Competing interests

The authors declare no competing interests.
