## [Peer Review File · Nature Communications]

Concentrated solar CO₂ reduction in H₂O vapour with >1% energy conversion efficiencyREVIEWER COMMENTS

Reviewer #1 (Remarks to the Author):

Photo-thermal reduction of CO₂ represents a promising method for sustainable fuel production. Here, Ren et al. report a specific Ni-CeO₂ catalysts accommodating high concentration of oxygen vacancies and Ni single-atoms, which gives high yield of CH₄ production with impressive energy-conversion efficiency of 1.14% and high selectivity of 100%. Besides, many techniques have been applied with the attempt to disclose the structure-performance relationship. Overall, these results seem interesting and may provide guidance for catalyst or system design in this field. However, several important issues should be adequately addressed.

1. The solar-to-chemical conversion efficiencies is an essential parameter for photocatalyst reactions. A benchmark material (or representative catalyst) for Photo-thermal CO₂ reduction should be tested to validate the reliability of reaction system and the credibility of the energy-conversion efficiency.
2. It is widely accepted that oxygen vacancies of CeO₂ are efficient sites to adsorb the reactants (e.g. CO₂ and H₂O). However, it is surprising that the role of oxygen vacancies is seldomly discussed in the reaction mechanism section. It is indicated that the adsorption energy of H₂O on the Ni-atom and oxygen vacancies should be compared.
3. Although various techniques were utilized, the organization of this manuscript is somewhat confusing, making it difficult to comprehensively gain the structure-performance relationship. The results and discussion part should be carefully improved.
4. (Line 199-202) It is found that the reflection peaks slightly shift to small-angle direction, which is assigned to excitation of electrons by the authors. This conclusion should be carefully double checked, since thermal expansion of the crystal cell can also induce such phenomenon.

Reviewer #2 (Remarks to the Author):

This manuscript utilized an oxygen-vacancy (V_o) rich CeO₂ catalyst with single atom Ni precisely anchored on its surface V_o sites to promote H₂O dissociation and achieve effective photothermal CO₂ reduction in H₂O vapour under concentrated light irradiation. The authors claimed the study should help further the development of H₂O-activating catalysts for CO₂ reduction and thereby expedite the practical utilization of STC technologies. Most of conclusions are well supported and the improved performance was investigated in details. Given its significance, timeliness and general interest to the broad readership of this journal, I would like to recommend this work be accepted after the following points are addressed:

1. The precise morphology of Ni in HRTEM poses a significant challenge. To overcome this limitation, the authors are encouraged to employ HAADF-STEM for a more detailed characterization, allowing for the labeling of individual Ni sites. Additionally, CO-DRIFT stands out as a valuable technique for

characterizing the nature of the metal active sites.

2. In Fig. 1 j-k, the diagrams depicting different atoms are conspicuously absent.

3. Accurate determination of nickel content is important. How can the ICP test of metal Ni exclude the influence of Ni foam?"

4. In the section on page 10, specifically lines 206-208, the statement 'which enhances the recombination between electrons and holes, thereby increasing the catalyst photo-thermal conversion efficiency' lacks clarity. From my perspective, the connection between electron-hole complexation and the photothermal conversion process is not well-established. It is recommended that the author provides a detailed explanation to enhance comprehension.

5. Mass spectrometry of CH₄ produced under 13CO₂ atmosphere should also be performed during the reaction process.

6. To determine the apparent activation energy accurately, it is essential to perform cases without light irradiation at the same temperature.

7. While the authors utilized thermocouples to monitor the catalyst's surface temperature, it is noted that this method did not provide an accurate representation of the actual temperature at the active center. The local temperature of Ni atoms may differ from the macroscopic temperature of the CeO₂ support. Therefore, the authors are advised to strive for a more precise, microscopic indication of temperature under irradiation.

8. The expression of Fig. S22 is ambiguous as to how the catalyst achieves different temperatures under constant concentrated solar irradiation. Is the temperature controlled by external heating? If so, how does this result in CH₄ yields lower than 192.75 μmol/cm²/h, which are only achieved under concentrated solar irradiation?

9. Further measurements of in-situ XPS are recommended to demonstrate changes in the valence states of both Ni and Ce.

Reviewer #3 (Remarks to the Author):

In this manuscript, the authors report a single-atom Ni anchored oxygen-vacancy (Vo) rich CeO₂ catalyst to achieve a CH₄ yield of 385.5 μmol/cm² with an energy conversion efficiency of 1.14% and a selectivity ~100% from photothermal CO₂ reduction in H₂O vapour under concentrated light irradiation. A lot of characterizations and theoretical calculations have been performed, and some results are interesting. However, the organization and writing of the manuscript is poor in whole, lacking a main line logically all through the manuscript. Following are some questions and comments.

1. As the authors mentioned (lines 71-73), CeO₂ has abundant Vo and lattice defects on its surface intrinsically. How did the authors obtain the NF@0.1%Ni@CeO₂ (less or no Vo) as a counterpart of NF@0.1%Ni@CeO₂-Vo?

2. While extensive XAS/EXAFS characterizations and DFT calculations are conducted, the coordination of Ni single atoms in CeO₂-Vo are not described clearly. In abstract, the authors claimed that "the single atoms Ni are precisely anchored on the surface Vo sites of CeO₂"; In the EXAFS characterization (lines 106, 1070, "These results suggest that the introduction of Vo causes surface reconstruction, with Ni replacing Ce and selectively occupying the Vo,"; In lines 110-111, the density functional theory (DFT)

(Figs. 1j, k) revealed that the single atom Ni loaded on the (111) face by replacing Ce, forming two structures with coordination numbers and bond lengths that align with the XAS results (Supplementary Table 3)."

--- What is the true case? If Ni replaces Ce, can it be taken as doping CeO₂ with Ni simply?

3. (Lines 95-97) "This can be attributed to the presence of V_o, which enables the transfer of electrons from O₂⁻ on the catalyst surface to Ni, reducing Ni to the +2 valence state;"

--- How could electrons reduce Ni to Ni²⁺? Is it possible? This is wrong in principle.

4. In Supplementary Figure 22, the yield of CH₄ increased with temperature, but the yield of CO remained unchanged. Why? How? Please explain.

5. In Supplementary Figure 29a, the PL intensity of NF@0.1%Ni@CeO₂-V_o was much lower than that of NF@0.1%Ni@CeO₂. But in Supplementary Figure 29b, the fitted lifetime (fluorescence decay time) of NF@0.1%Ni@CeO₂-V_o was longer than that of NF@0.1%Ni@CeO₂. This is incorrect conceptually. The authors are suggested to re-check by studying some papers on the fundamentals of photoluminescence (e.g., Angew. Chem. Int. Ed. 2023. 62, e202313537, doi.org/10.1002/anie. 202313537). The nonradiative/radiative decays and their respective contributions to catalytic activity should be clarified.

6. (Lines 150-152) "However, the impurity state of the d orbital is located at the top of the VBM, which implies that the VBM hole capture is equivalent to a relaxation process with an energy level that is similar to the bandgap value between the CBM and VBM, increasing the possibility of capturing the photo-generated holes."

--- How could an energy level between the CBM and VBM capture the photo-generated holes? Please explain more clearly.

7. (Lines 204-207) "In situ electron paramagnetic resonance (EPR) also provided compelling evidence for the increase in V_o in the catalysts under concentrated solar irradiation conditions (Fig. 2h), which enhances the recombination between electrons and holes, thereby increasing the catalyst photothermal conversion efficiency.

--- How could the enhancement of recombination between electrons and holes increase the catalyst photothermal conversion efficiency?

8. (Lines 381-383) "The results showed that the C in CO₂ cannot coordinate to the Ni sites of the catalyst surface because they are positively charged. Therefore, CO₂ is not stably adsorbed to the surface by bonding single-atom Ni." If this is the case, is any other single-atom catalyst not favorable for the adsorption/activation of CO₂ molecules? The fact is that it is not true.

9. (Lines 412-413) "Any remaining O species may form O₂ molecules with other O species or fill the V_o, leading to Ni deactivation, which could shift from the planar quadrilateral coordination structure to an octahedral one."

-- Did the authors detect evolution of O₂? Otherwise, Ni deactivation would occur since the O species will fill the V_o, as the authors say.

10. In the session "Mechanism of light and thermal effects over the Ni single atom", the authors intended to prove the photo-thermal coupled CO₂ reduction mechanism. But they did not design and conduct appropriate comparative experiments as many previously reported research works on photothermal CO₂ reduction did to clarify the respective effects.

11. Regarding the roles of Ni single atom and V_o, the authors seem not to give consistent descriptions that are very confusing to the reviewer. 1) "The CO₂ temperature-programmed desorption spectra

shown in Supplementary Fig. 28 confirm that Vo and Ni can provide surface basic sites to facilitate the CO₂ adsorption and decomposition.” 2) The in situ DRIFTS show that “The remarkably strong m-CO₃2- peaks indicate that CO₂ is adsorbed on the lattice O of CeO₂ instead of on Ni.”

--- What is the true case? Please give more explanations.

12. Some experimental details. 1) The light source spectra and intensities of both the concentrated solar irradiation and the non-concentrated solar irradiation should be provided. 2) The isotope test should be performed to confirm the origin of CH₄ is from CO₂ but not from other possible carbon-containing sources and impurities. 3) The authors used the unit of $\mu\text{mol cm}^{-2}$ or $\mu\text{mol cm}^{-2} \text{ h}^{-1}$ for evaluating the catalytic activity. However, most previously reported works used the unit of $\mu\text{mol g}_{\text{cat}}^{-1} \text{ h}^{-1}$. In the Supplementary table 5, the comparison of the authors' catalyst with others would not be appropriate.

13. The reviewer's personal taste. Using concentrated solar irradiation to improve the catalytic efficiency is more likely a technical skill but not a scientific merit.

Given the aforementioned questions and comments, I could not recommend acceptance of the current paper.

Point-by-Point Responses to Reviewers' Comments

Reviewer #1:

“1. Photo-thermal reduction of CO₂ represents a promising method for sustainable fuel production. Here, Ren et al. report a specific Ni-CeO₂ catalysts accommodating high concentration of oxygen vacancies and Ni single-atoms, which gives high yield of CH₄ production with impressive energy-conversion efficiency of 1.14% and high selectivity of 100%. Besides, many techniques have been applied with the attempt to disclose the structure-performance relationship. Overall, these results seem interesting and may provide guidance for catalyst or system design in this field. However, several important issues should be adequately addressed.”

We would like to sincerely thank this reviewer for the positive feedback on our manuscript. According to the following up comments and constructive suggestions, we firstly compared the CO₂ reduction activities of representative photothermal catalysts including TiO₂, WO₃, and ZrO₂ to verify the reliability of the reaction system and the credibility of the energy conversion efficiency. To investigate the role of oxygen vacancies (V_o) in the reaction mechanism, we conducted the temperature-programmed desorption of H₂O (H₂O-TPD) tests and DFT theoretical calculations to compare the adsorption energies of H₂O on Ni and V_o. In addition, we have meticulously improved the results and discussion sections by aligning them with the 'structure-property' relationship and we have also included comprehensive characterizations of NF@0.1%Ni@CeO₂-V_o samples. These additional contents have been incorporated into the revised main text and/or the Supplementary Information file.

“2. The solar-to-chemical conversion efficiencies is an essential parameter for photocatalyst reactions. A benchmark material (or representative catalyst) for Photo-thermal CO₂ reduction should be tested to validate the reliability of reaction system and the credibility of the energy-conversion efficiency.”

We would like to sincerely thank this reviewer for this professional comment.

In principle, the solar-to-chemical (STC) energy conversion efficiency could be obtained by comparing input energy and output energy, as shown in the following formula (*Renewable Sustainable Energy Rev.*, 2024, **189**, 114053):

$$\eta_{STC} = \frac{qm}{IA t} \times 100\% \left(\frac{J \cdot mol^{-1} \cdot mol}{W \cdot cm^{-2} \cdot cm^2 \cdot s} = \frac{J}{W \cdot s} = 1 \right) \quad (S1)$$

Where q is the combustion heat value of CO and CH₄ ($q_{CO} = 283.5 \text{ kJ} \cdot \text{mol}^{-1}$, $q_{CH_4} = 890.8 \text{ kJ} \cdot \text{mol}^{-1}$), m denotes the mole of produced CH₄ (mol), I denotes the light intensity ($W \cdot \text{cm}^{-2}$), A is the irradiation area (cm^2), and t denotes the reaction time (s).

Much of previous research has reported the method of calculating the energy conversion efficiency and realized efficient photocatalytic reaction by concentrated light technology.

Please refer to the literature cited below:

1. *Chem. Eng. J.*, 2023, **468**, 143831.
2. *Ind. Eng. Chem. Res.*, 2023, **62**, 9463–9473.
3. *Chem. Eng. J.*, 2024, **479**, 147529.
4. *Angew. Chem. Int. Ed.*, 2023, **62**, e202313868.
5. *ACS Nano*, 2023, **17**, 10976–10986.
6. *Science*, 2010, **330**, 1797–1801.
7. *Adv. Energy Mater.*, 2023, **13**, 2300071.
8. *ACS Catal.*, 2023, **13**, 15591–15602.

According to the valuable suggestion, we examined representative catalysts (including WO₃, ZrO₂, and TiO₂) to verify the reliability of the reaction system and the credibility of the energy conversion efficiency. XRD patterns and TEM images demonstrate that the prepared materials are in their typical phase states (Supplementary Figs. 25, 26).

Supplementary Fig. 27 shows the catalytic performance of CO₂ reduction under different conditions. The corresponding STC energy conversion efficiencies were calculated based on CO and CH₄ yields. For WO₃, ZrO₂, TiO₂, the corresponding STC energy conversion efficiency increased from 0.014%, 0.017%, 0.011% under non-concentrated solar irradiation to 0.073%, 0.086%, 0.047% under concentrated solar irradiation, respectively. This result represents a 5.2,

5.1, and 4.3 times improvement induced by concentrated solar irradiation. The energy conversion efficiency of this work is acceptable when compared to other studies under similar conditions (Supplementary Table 5). This indicates that the current reaction and the calculation of STC energy conversion efficiency are scientifically sound and meets the standards of the field. These results are involved in the revised manuscript and are also attached below:

Supplementary Fig. 25 | XRD patterns of different catalysts.

Supplementary Fig. 26 | a, b, and c TEM image of WO₃ (a), ZrO₂ (b), and TiO₂ (c) catalysts. d, e, and f HRTEM image of WO₃ (d), ZrO₂ (e), and TiO₂ (f) catalysts.

Supplementary Fig. 27 | Photothermal CO₂ reduction performance of WO₃, ZrO₂, and TiO₂ catalysts under different conditions.

Supplementary Table 5 | Comparison of energy conversion efficiencies of photo and photothermal catalysis.

Catalysts	Light Intensity (mW cm ⁻²)	Reaction medium	Reduction products	Catalyst dosage (mg)	System temperature (K)	Yield	$\eta_{\text{STC}}(\%)$
NF@0.1%Ni@CeO ₂ -V _o (This work)	4200	CO ₂ and H ₂ O vapour	CH ₄	1	635.15	192.75 $\mu\text{mol cm}^{-2} \text{h}^{-1}$	1.14
WO ₃	4000	CO ₂ and H ₂ O vapour	CO and CH ₄	1	511.15	26.51 and 3.36 $\mu\text{mol}\cdot\text{cm}^{-2}\cdot\text{h}^{-1}$	0.073
ZrO ₂	4000	CO ₂ and H ₂ O vapour	CO and CH ₄	1	529.15	20.63 and 7.36 $\mu\text{mol}\cdot\text{cm}^{-2}\cdot\text{h}^{-1}$	0.086
TiO ₂	4000	CO ₂ and H ₂ O vapour	CO and CH ₄	1	485.15	15.42 and 2.65 $\mu\text{mol}\cdot\text{cm}^{-2}\cdot\text{h}^{-1}$	0.047
Au-Mt-H6 ²	720	CO ₂ and H ₂ O	CO and CH ₄	10	454.15	0.14 and 0.17 μmol	0.000037

TF@TNT/0.4CoO _x - 0.1CuO ³	4266	vapour CO ₂ and H ₂ O	CO and CH ₄	1	573.15	m ⁻² 8.45 and 58.2 μmol cm ⁻² h ⁻¹	0.35
Bi ₂ S ₃ @In ₂ S ₃ ⁴	1150	vapour CO ₂ and H ₂ O	CO and C ₂ H ₄	5	513.15	1.95 and 11.81 μmol g ⁻¹ h ⁻¹	0.00033
Pt/ZnO ⁵	4200	vapour CO ₂ and H ₂ O	CO and CH ₄	20	540.15	115.32 and 8 μmol g ⁻¹ h ⁻¹	0.264
Au _x /TJU-16 ⁶	300 W Xe lamp	vapour CO ₂ and H ₂ O	CO and CH ₄	40	/	10.1 and 2.5 μmol g ⁻¹ h ⁻¹	0.034
Au@Cr ₂ O ₃ ⁷	1600	vapour CO ₂ and H ₂ O	CO and H ₂	1000	/	0.074 and 0.013 mol g ⁻¹ h ⁻¹	0.89
CuO _x /TiO ₂ ⁸	300 W Xe lamp	vapour CO ₂ and H ₂ O	CO and CH ₄	50	/	0.32 and 2.29 μmol g ⁻¹ h ⁻¹	0.0372
Sn/In ₂ O ₃ ⁹	300 W Xe lamp	vapour CO ₂ and H ₂ O	CO	50	/	17.9 μmol g ⁻¹ h ⁻¹	0.05
Rh/Al nanoantenna ¹⁰	11300	CO ₂ and H ₂	CH ₄ and H ₂ O	20	973.15	550 mmol g ⁻¹ h ⁻¹	0.0011
Co/Al ₂ O ₃ ¹¹	3295	CO ₂ and H ₂	CH ₄	/	803.15	6036 μmol g ⁻¹ h ⁻¹	0.0002

...

The energy conversion efficiency is a crucial indicator for evaluating photocatalytic activity. Three benchmark photothermal catalysts were synthesized for comparison with the present catalyst and existing studies (WO₃, ZrO₂, and TiO₂, see Supplementary Figs. 25, 26 the structure/phase information). The direct STC energy conversion of present catalyst exhibits the highest efficiency (1.14%; Fig. 2f; Supplementary Fig. 27; Supplementary Table 5), with a CH₄ selectivity of approximately 100% (Supplementary Fig. 28; Supplementary Table 6)...

1.2.3. Preparations of WO₃

1 mmol Na₂WO₄·2H₂O along with nickel foam (1 cm × 1 cm × 2 mm) were dissolved in 20 mL H₂O and the solution was then stirred for 1 h. Next, 0.5 mmol N₂H₄·H₂O was added to the solution and stirred for 1 h. The pH of the mixture solution was then adjusted to 3 using concentrated hydrochloric acid and the solution was stirred for an additional 0.5 h. Afterward, the mixture solution was heated at 180 °C for 12 h. Finally, the mixture solution was centrifuged, washed, and dried, which was labeled as WO₃.

1.2.4. Preparations of ZrO₂

We dissolved 8 g of ZrOCl₂·8H₂O and nickel foam (1 cm × 1 cm × 2 mm) in 70 mL of deionized water and stirred the solution for 30 min at 600 r/min. Next, we transferred the solution to a PTFE-lined reactor, sealed it, and heated it to 150 °C for 6 h. After cooling the samples to room temperature, we separated the product by centrifugation, washed it three times with deionized water, and dried it in a vacuum oven at 70 °C for 10 h. This sample was labeled as ZrO₂.

1.2.5. Preparations of TiO₂

A mixture of 1.5 mL of HF solution (43% wt) and 15 mL of C₁₆H₃₆O₄Ti solution (99.5% wt) was stirred for 1 h. Subsequently, a solvothermal reaction was carried out at 200 °C for 24 h in a 25 mL Teflon autoclave. After natural cooling to room temperature, the precipitants were collected and fully washed with distilled deionized H₂O, then dried in a vacuum oven at 70 °C for 10 h. Then, the powder and nickel foam (1 cm × 1 cm × 2 mm) were stirred in deionized water for 1 h and then aged for 2 h. Finally, the TiO₂ nanosheets were annealed for 3 h in air atmosphere at 450 °C with the temperature elevation rate of 5 °C/min.

“3. It is widely accepted that oxygen vacancies of CeO₂ are efficient sites to adsorb the reactants (e.g. CO₂ and H₂O). However, it is surprising that the role of oxygen vacancies is seldomly discussed in the reaction mechanism section. It is indicated that the adsorption energy of H₂O on the Ni-atom and oxygen vacancies should be compared.”

We would like to thank the reviewer for this pertinent comment. To further discuss the role of

V_o in the reaction mechanism, we performed H_2O -TPD tests and DFT theoretical calculations to compare H_2O adsorption energies on Ni single atoms and V_o .

Due to the high-temperature environment on the catalyst surface under concentrated solar irradiation, the enhancement in H_2O -catalyst interaction is crucial for improving the catalyst activity. We verified the H_2O -TPD on the following catalysts: $NF@CeO_2$, $NF@CeO_2-V_o$, and $NF@0.1\%Ni@CeO_2-V_o$. The results demonstrate that the introduction of both Ni single atoms and V_o enhances the dissociation capacity of H_2O (Supplementary Fig. 59). Furthermore, according to DFT calculation results shown in Supplementary Fig. 57, the adsorption energies of H_2O on CeO_2-V_o and $Ni@CeO_2$ are -0.67 eV and -1.47 eV, respectively. It is worth noting that on CeO_2-V_o , H_2O is adsorbed by the unsaturated Ce atoms instead of V_o sites. Meanwhile, on $Ni@CeO_2$, H_2O is adsorbed by the Ni single atoms instead of Ce atoms (Supplementary Fig. 58). Taken together, in this system, V_o does not directly participate in H_2O dissociation. Similar phenomena can also be observed in the process of water adsorbed on other single-atom catalysts loaded on metal oxide materials (*Angew. Chem. Int. Ed.*, 2022, **134**, e202203249).

The results and discussion are involved in the revised manuscript and are also attached below.

Supplementary Fig. 57 | Adsorption energies of H_2O on V_o site and Ni SAC site.

Supplementary Fig. 58 | Adsorption structure of H₂O on V_o site and Ni SAC site.

Supplementary Fig. 59 | a, b, and c TPD spectra from water adsorbed on NF@CeO₂ (a), NF@CeO₂-V_o (b), NF@0.1%Ni@CeO₂-V_o (c).

...However, H₂O molecules are dissociated upon applying an intense laser field. Owing to the presence of captured photo-generated holes on the H₂O molecules, the released O species are rapidly desorbed from the Ni sites (Fig. 5f). H species are then transferred to CO₂ and captured by an O atom, leading to CO₂ reduction and completing the methanation process. The adsorption energies of H₂O on V_o and Ni single atoms are -0.67 eV and -1.47 eV, respectively (Supplementary Fig. 57). It is worth noting that on CeO₂-V_o, H₂O is adsorbed by the unsaturated Ce atoms instead of V_o sites. Meanwhile, on Ni@CeO₂, H₂O is adsorbed by the Ni single atoms instead of Ce atoms (Supplementary Fig. 58). Additionally, H₂O-TPD demonstrates that the incorporation of Ni single atoms and V_o can significantly enhance the dissociation capacity of H₂O (Supplementary Fig. 59)...

“4. Although various techniques were utilized, the organization of this manuscript is somewhat confusing, making it difficult to comprehensively gain the structure-performance relationship. The results and discussion part should be carefully improved.”

We are grateful for this valuable comment. Careful changes have been made to the results and discussion sections, using the 'structure-performance' relationship as the logical thread. These changes aim to improve the comprehensibility and logical structure of the text while maintaining clear and objective language. The revised manuscript and associated Supplementary Information now include additional content, which is distinguished by a yellow background.

The manuscript now comprises **five sections**. The part of the '**Structural characterization of catalysts**' details the morphological and coordination characterization of NF@0.1\%Ni@CeO_2 and $\text{NF@0.1\%Ni@CeO}_2\text{-V}_o$ catalysts, along with the coordination environment of the single-atom Ni. In the second subsection '**Photo-thermal catalytic performance**', the effects of different Ni single atom coordination environments on photothermal catalytic performance are discussed. The third subsection '**Mechanism of light and thermal effects over the Ni single atom**' summarizes how concentrated solar irradiation induces light and thermal effects that impact the mechanism of single atom Ni catalysis. In the fourth subsection '**Charge carrier dynamics**', the carrier migration and separation are summarized. Finally, in the fifth subsection '**CO₂ reduction mechanism under concentrated solar irradiation**', the mechanism of catalysed CO₂ reduction is verified *via* molecular dynamics of surface excited states and relevant in situ characterisation. Among them, the sequence of '**Charge carrier dynamics**' and '**Mechanism of light and thermal effects over the Ni single atom**' has been rearranged to achieve a clearer logical structure. The logical diagram illustrates the overall framework of the manuscript (Fig. R1). Please refer to the highlighted changes in yellow in the Results and Discussion and Supporting Information sections.

Fig. R1 | General framework of manuscript.

“5. (Line 199-202) It is found that the reflection peaks slightly shift to small-angle direction, which is assigned to excitation of electrons by the authors. This conclusion should be carefully double checked, since thermal expansion of the crystal cell can also induce such phenomenon.”

We would like to thank the reviewer for this insightful comment. We also agree that typical oxide materials may undergo thermal expansion at high temperatures, causing a change in the lattice constant and resulting in a shift in the position of the diffraction peak. To exclude the possibility of thermal expansion induced peak shift, in-situ XRD tests were conducted on the NF@0.1%Ni@CeO₂-V_o catalyst at various temperatures (Supplementary Fig. 34). The diffraction peak position did not shift towards the small-angle direction as the temperature increased from 373.15K to 673.15K. In fact, thermal expansion induces uniform lattice distortions and the distance changes between each crystal plane. However, the changes in lattice constants induced by thermal effects are limited, so the shifts in the positions of the diffraction peaks are weak (*J. Phys. Chem. C*, 2010, **114**, 15123–15128; *J. Am. Ceram. Soc.*, 2009, **92**, 2713–2717; *Mineral. Geochem.*, 2003, **54**, 1–112). On the contrary, photoexcitation could

induce observable diffraction peak changes. When exposed to high flux photon density, the electrons in CeO₂ will be excited to higher energy levels. The electron excitation causes a change in the oxygen ion position within the lattice and thereby affects the diffraction peak position (*J. Phys. Chem. C*, 2008, **112**, 18769–18773; *J. Mater. Sci.*, 2017, **52**, 4959–4970; *J. Appl. Phys.*, 2009, **106**, 123525). In conclusion, the small-angle shift of the diffraction peaks is mainly attributed to the diverse lattice distortions caused by high flux photon density excitation. The results and discussion are involved in the revised manuscript and are also attached below.

Supplementary Fig. 34 | In-situ XRD pattern of NF@0.1%Ni@CeO₂-V_o under different temperatures.

...Furthermore, we conducted in situ XRD tests, finding that the catalyst diffraction peak shifted towards the small-angle direction under concentrated solar irradiation. This phenomenon can be attributed to two reasons: 1) thermal expansion and 2) high flux photon density. The variable-temperature XRD patterns exhibit no shift of the diffraction peaks with increasing temperature. In fact, thermal expansion induces uniform lattice distortions and the distance changes between each crystal plane. However, the changes in lattice constants induced by thermal effects are limited, so the shifts in the positions of the diffraction peaks are weak (Supplementary Fig. 34). On the other hand, high flux photon densities can excite electrons in the catalyst, causing the lattice atoms to leap to the defect sites. These leaps can lead to the formation of defects, which can alter the distances between atoms in the crystal, resulting in a change in the lattice constant and shift of the diffraction peak (Fig. 2g)...

Reviewer #2

“1. This manuscript utilized an oxygen-vacancy (V_o) rich CeO_2 catalyst with single atom Ni precisely anchored on its surface V_o sites to promote H_2O dissociation and achieve effective photothermal CO_2 reduction in H_2O vapour under concentrated light irradiation. The authors claimed the study should help further the development of H_2O -activating catalysts for CO_2 reduction and thereby expedite the practical utilization of STC technologies. Most of conclusions are well supported and the improved performance was investigated in details. Given its significance, timeliness and general interest to the broad readership of this journal, I would like to recommend this work be accepted after the following points are addressed.”

We are grateful to the reviewer for the overall positive feedback on our work and for the valuable suggestions and comments. We have diligently addressed each of these comments in a point-by-point manner and have incorporated additional contents into the revised manuscript and corresponding Supplementary Information.

“2. The precise morphology of Ni in HRTEM poses a significant challenge. To overcome this limitation, the authors are encouraged to employ HAADF-STEM for a more detailed characterization, allowing for the labeling of individual Ni sites. Additionally, CO-DRIFT stands out as a valuable technique for characterizing the nature of the metal active sites.”

We would like to thank this reviewer for this valuable comment. High-angle annular dark-field scanning transmission electron microscopy (HAADF-STEM) has been employed accordingly to examine the precise morphology of Ni involved in the $NF@0.1\%Ni@CeO_2-V_o$ catalyst (see revised Fig. 1). Clearly, the lattice spacing of 0.31 nm corresponds to the (111) crystallographic plane of CeO_2 (Fig. 1b). The presence of Ni single atoms can also be clearly observed (Fig. 1c).

In addition, the CO-DRIFT was employed to verify the nature of the metal active sites, accordingly to the reviewer's suggestion. In principle, if it is a single active centre (single atom), CO will be linearly adsorbed, while for a multi-active centre (cluster or particle), it is bridged adsorbed. The state of the metal can be determined at $1800-2400\text{ cm}^{-1}$ based on the observation of whether CO is bridged or linear adsorbed. As shown in Supplementary Fig. 5a for Ni

nanoparticles, firstly the amount of adsorption increases as the CO adsorption time increases, but the process involves both chemisorption and physisorption. The catalytic activity is mainly affected by chemisorption, so it is necessary to use N₂ purging to eliminate the effect of physical adsorption. After N₂ purging, no obvious characteristic peaks appear in the sample, which can be attributed to the poor chemisorption capacity of the base metal Ni itself for CO. Similar phenomenon could be also observed on the NF@0.1%Ni@CeO₂-V_o catalyst (Supplementary Fig. 5b). It therefore could be concluded that for Ni, CO-DRIFT cannot effectively distinguish whether it is in the form of single atom or not. In fact, CO-DRIFT was commonly used to verify the single-atom nature of the catalysts containing noble metals (Pd, Au, Pt, Rh, etc.), which have a strong chemisorption capacity for CO (*Nat. Commun.*, 2021, **12**, 1549; *Science*, 2014, **346**, 1498; *Small Struct.*, 2023, **4**, 2200115; *Angew. Chem. Int. Ed.*, 2021, **60**, 24220). In addition, the CO-TPD results indicate that the NF@0.1%Ni@CeO₂-V_o catalyst has poor chemisorption and desorption of CO, consistent with the above observations (Supplementary Fig. 6). We have included the results in the revised manuscript. Please also find below the revision.

Fig. 1 | Morphological characterizations of NF@0.1%Ni@CeO₂-V_o. b, c HAADF-STEM images of NF@0.1%Ni@CeO₂-V_o.

Supplementary Fig. 5 | a, b In-situ CO-DRIFT images of Ni NPs@ γ -Al₂O₃ samples (a), and NF@0.1%Ni@CeO₂-V_o samples (b).

Supplementary Fig. 6 | CO-TPD images of NF@0.1%Ni@CeO₂-V_o.

...In-situ CO-DRIFT images failed to identify whether Ni is a single atom due to the poor chemisorption of CO on non-precious metal Ni (Supplementary Figs. 5, 6). HAADF-STEM images of NF@0.1%Ni@CeO₂-V_o show lattice stripes with a crystal plane spacing of 0.31 nm, corresponding to the (111) crystal plane of CeO₂ on the primarily exposed planes (Fig. 1b). Besides, the presence of Ni single atoms was observed (Fig. 1c).

“3. In Fig.1 j-k, the diagrams depicting different atoms are conspicuously absent.”

We would like to thank this reviewer for proof-reading of the figures. Fig. 1, j and k, have been

revised with the different atoms clearly been labeled. Please see Fig. 1j and 1k.

Fig. 1 | Morphological characterizations of NF@0.1%Ni@CeO₂-V_o. **j, k Ni–O coordination environment in NF@0.1%Ni@CeO₂ (**j**) and NF@0.1%Ni@CeO₂-V_o catalysts (**k**).**

“4. Accurate determination of nickel content is important. How can the ICP test of metal Ni exclude the influence of Ni foam?”

We would like to thank the reviewer for the pertinent comments on our manuscript. As mentioned in 1.2.2 of the Method section, the NF@0.1%Ni@CeO₂-V_o catalyst was prepared by growing it on nickel foam. It is important to note that after the hydrothermal reaction, there is residual catalyst powder in the reaction liner. The catalyst powder was cleaned three times with ethanol and water, then dried under vacuum at 70 °C for 8 h. To guarantee accuracy, we compared the performance of the catalyst grown on nickel foam with that of the residual catalyst powder in a CO₂ reduction test. As shown in Supplementary Figs. 23a, b, no notable difference in the catalytic activity or CH₄ selectivity could be found. Consequently, the role of nickel foam is only as a support material of the catalyst and does not participate in the catalytic reaction. In this case, ICP test was only implemented to the catalyst powder to calibrate the Ni content. Please also find below the revision to the manuscript.

Supplementary Fig. 23 | a, b Photothermal catalytic CH₄ yields from NF@0.1%Ni@CeO₂-V_o and 0.1%Ni@CeO₂-V_o powder catalysts under different conditions (non-concentrated solar irradiation represents 420 mW/cm², and concentrated solar irradiation represents 4200 mW/cm²) (a) and CH₄ selectivity of NF@0.1%Ni@CeO₂-V_o and 0.1%Ni@CeO₂-V_o powder catalysts (b).

...Under identical conditions, catalytic activities of 0.1%Ni@CeO₂-V_o catalysts loaded on nickel foam did not present notable difference from those of 0.1%Ni@CeO₂-V_o powder catalysts (Supplementary Figs. 23a, b). This result demonstrates that nickel foam does not actively participate in the reaction. Therefore, it is rational to consider nickel foam as a support material for the catalyst...

“5. In the section on page 10, specifically lines 206-208, the statement ‘which enhances the recombination between electrons and holes, thereby increasing the catalyst photo-thermal conversion efficiency’ lacks clarity. From my perspective, the connection between electron-hole complexation and the photothermal conversion process is not well-established. It is recommended that the author provides a detailed explanation to enhance comprehension.”

We would like to thank the reviewer for the professional comments. This statement is intended to indicate the enhancement of heat utilization efficiency by Ni single atoms and oxygen vacancies (V_o). As illustrated in Fig. 5h, photo-generated carriers can be captured by Ni single atoms and release high-frequency phonons *via* recombination. Meanwhile, Ni single atoms and

V_o form atomic-scale active sites. These active sites, combined with hot spot effect induced by carriers' recombination, promote the CO_2 reduction reaction. The surface temperature elevation with the introduction of Ni single atoms and V_o is shown in Supplementary Fig. 35a. According to the fluorescence spectrum measurement results of $NF@CeO_2$ and $NF@CeO_2-V_o$ materials in Supplementary Figs. 35b, c, V_o indeed possess the capability to capture photo-generated charges. On the other hand, the photogenerated electrons and holes are spatially separated in the surface depletion layer, leading to lower PL intensity and longer carrier lifetime. Please also find below the revision to the manuscript.

Supplementary Fig. 35 | a Surface temperature of different catalysts under concentrated light conditions. **b** PL spectra measured and for $NF@CeO_2$ and $NF@CeO_2-V_o$. **c** Time-resolved transient PL decay spectra.

...In situ electron paramagnetic resonance (EPR) also provided compelling evidence for the increase in V_o in the catalysts under concentrated solar irradiation conditions (Fig. 2h), which enhances the recombination between electrons and holes. Photogenerated carriers can be captured by Ni single atoms and release high-frequency phonons *via* recombination. Meanwhile, Ni single atoms and V_o form atomic-scale active sites. These active sites, combined with hot spot effect induced by carriers' recombination, promote the CO_2 reduction reaction. The surface temperature elevation with the introduction of Ni single atoms and V_o is shown in Supplementary Fig. 35a. Besides, V_o indeed possess the capability to capture photo-generated charges (Supplementary Figs. 35b, c)...

“6. Mass spectrometry of CH_4 produced under $^{13}CO_2$ atmosphere should also be performed during the reaction process.”

We would like to thank the reviewer for this comment. $^{13}\text{CO}_2$ isotope labeling experiments were carried out under the same conditions. The gas products were tested by gas chromatography-mass spectrometry (GC-MS, Agilent 8860). In the experiment, helium gas was used as a carrier gas, and sample injection temperature and auxiliary heating temperature were both set at 125 °C, with the constant flow rate being 1.2 mL·min⁻¹. The initial temperature of the column box is 60 °C, and the temperature is heated to 180 °C at 10 °C·min⁻¹, and the storage temperature is kept for 1 min. As shown in Supplementary Fig. 24, the products were detected by gas chromatography-mass spectrometry (GC-MS). The main signal at $m/z = 17$ is assigned to $^{13}\text{CH}_4$ and the other signal at $m/z = 29$ can be assigned to ^{13}CO . This result indicates that the CO and CH₄ generation originates from the CO₂ reactant, thereby ruling out catalyst decomposition or carbon pollution as contributing factors. Please refer to Supplementary Fig. 24 and find below the revision to the main text.

Supplementary Fig. 24 | $^{13}\text{CO}_2$ isotopic tracing experiments of NF@0.1%Ni@CeO₂-V_o.

...To identify the carbon source of the reduced products, isotope tracing experiments were carried out using reactant $^{13}\text{CO}_2$. The main signal at $m/z = 17$ is assigned to $^{13}\text{CH}_4$ and the other signal at $m/z = 29$ can be assigned to ^{13}CO . This result indicates that the CO and CH₄ generation originates from the CO₂ reactant, thereby ruling out catalyst decomposition or carbon pollution as contributing factors (Supplementary Fig. 24).

...

“7. To determine the apparent activation energy accurately, it is essential to perform cases without light irradiation at the same temperature.”

We would like to thank the reviewer for this valuable comment. The operating temperature for thermal catalytic conversion of CO₂ into CO or CH₄ is generally beyond 1000 °C (*Prog. Energy Combust. Sci.* 2019, **74**, 1–30). The catalyst temperature in this study is 362.1 °C, significantly lower than the temperature aforementioned.

According to the reviewer’s comment, we performed the experiments under dark conditions at 200 °C-400 °C using an external heating device (see Fig. 2d). Our findings indicate that no CH₄ released throughout the process. However, exposure to the light irradiation of 1200 mW/cm² at the same temperature leads to notable CH₄ production. It is therefore rational to conclude that the reaction is not dominated by thermal catalysis.

In addition, the surface temperature of the catalysts was tested at controlled temperatures with consistent light irradiation intensity (Supplementary Table 9). This allowed for the calculation of the apparent activation energy using formula (1):

$$\ln k = \ln A - E_a/RT \quad (1)$$

where k, R, T, E_a and A denote the rate constant, molar gas constant, thermodynamic temperature, apparent activation energy, and frequency factor, respectively. Besides, **since the yields of CO₂ to CH₄ were not detectable in the absence of light, this formula is not applicable to the calculation of the apparent activation energy without light irradiation at the corresponding temperature.** However, these findings imply that the apparent activation energy is sufficiently high in the absence of light irradiation. Otherwise, the conversion of CO₂ to CH₄ would be observable.

Fig. 2 | d Yields of CH₄ from the NF@0.1%Ni@CeO₂-V_o catalyst in the dark and with the addition of 1200 mW/cm² solar light at different temperatures.

Supplementary Table 9 | Parameters for CH₄ production under concentrated solar irradiation and non-concentrated solar irradiation.

1000/T (K ⁻¹)	lnk _{CH₄} yield (μmol/cm ² /h)	1000/T (K ⁻¹)	lnk _{CH₄} yield (μmol/cm ² /h)
1.57	5.89	2.62	1.59
1.68	5.76	2.7	1.49
1.81	5.68	2.8	1.119
1.93	5.6	2.92	0.89
2.06	5.56	3.02	0.66

“8. While the authors utilized thermocouples to monitor the catalyst's surface temperature, it is noted that this method did not provide an accurate representation of the actual temperature at the active center. The local temperature of Ni atoms may differ from the macroscopic temperature of the CeO₂ support. Therefore, the authors are advised to strive for a more precise, microscopic indication of temperature under irradiation.”

We would like to thank the reviewer for this suggestion. It is a very important issue to accurately measure temperature of the active center. However, measurement of the temperature at the atomic scale is still a challenging task. Currently, available methods for temperature measurement at microscale mainly relies on the utilization of infrared thermometer or a micro-sized thermocouple, while the spatial resolution is restricted in the μm scale. Recently, in-situ scanning Raman microscopy has been demonstrated with atomic-level temperature determination. Unfortunately, currently we don't have condition to conduct such experiments. Although accurately measuring the temperature of atomic sites is important to this study, our understanding toward the photothermal coupling mechanism at the atomic level is achieved through current characterization and computational calculation. Anyway, it is an interest and challenging subject that deserves further research.

“9. The expression of Fig. S22 is ambiguous as to how the catalyst achieves different temperatures under constant concentrated solar irradiation. Is the temperature controlled by

external heating? If so, how does this result in CH₄ yields lower than 192.75 μmol/cm²/h, which are only achieved under concentrated solar irradiation?”

We would like to thank this reviewer for the meticulous examination of our manuscript. We are also sorry for this potentially misleading statement. **In our experiment, no external heat source was applied.** The *x*-axis of Supplementary Fig. 22 (current Supplementary Fig. 38) in the original manuscript represents the ambient temperature of the reactor cavity during catalysis, rather than the surface temperature of the catalyst. The catalytic process is cooled by introducing circulating water (0 °C/30 °C) at a controlled flow rate. The temperature in the chamber decreases as the flow rate increases, resulting in faster heat loss (Supplementary Fig. 37). The effect of ambient temperature on the catalyst was detected using thermocouples, revealing a direct correlation between rising ambient temperatures and an increase in CH₄ yield. The objective is to confirm that the catalytic system operates primarily through heat-assisted photocatalysis.

To avoid misleading, we have made following changes to the referred figures and table by (1) including an error bar in Supplementary Fig. 38 and a revision to the figure caption; (2) indicating in Supplementary Table 10 the temperature of the water, the ambient temperature, and the flow rate of the water. Please also find below the revision.

Supplementary Fig. 37 | Physical diagram of the catalytic process.

Supplementary Fig. 38 | CO₂ reduction performance of the NF@0.1%Ni@CeO₂-V_o catalyst at different temperatures under concentrated solar irradiation conditions. The temperature refers to the ambient temperature of the reactor cavity during catalysis.

Supplementary Table 10 | Experimental details parameters.

Cooling circulating water temperature (°C)	Temperature of the inner chamber (°C) ^a	Water circulation flow rate (mL/min)
0	50	600
0	100	240
30	150	600
30	200	240
/	235	/

^aTemperature tested by thermocouple.

...This was confirmed by determining the CH₄ yield at different temperatures under a constant concentrated solar irradiation intensity (Supplementary Figs. 37, 38; Supplementary Tables 10, 11). The CH₄ yield increased with increasing temperature, demonstrating the auxiliary role of heat in the catalysis...

“9. Further measurements of in-situ XPS are recommended to demonstrate changes in the valence states of both Ni and Ce.”

We would like to thank the reviewer for this valuable suggestion. Yes, the valence states of Ni

and Ce are essential for the study of catalytic processes. Supplementary Figure 36a shows the in-situ XPS spectra of the Ni 2p orbitals of the NF@0.1%Ni@CeO₂-V_o catalyst after irradiation with the reactants CO₂ and H₂O vapour for 120 min. However, the spectra only show the satellite peak at 881.04 eV. The signal intensity of Ni is very weak due to its low content (0.1%). Therefore, we employed the XAS spectra to capture the valence state of Ni on the NF@0.1%Ni@CeO₂-V_o catalyst before and after the photothermal reaction. The result indicates that the valence states of Ni was not changed (Supplementary Fig. 30a).

In the revised manuscript, we added in-situ XPS to check the valence changes in the element Ce (Supplementary Fig. 36b). The Ce 3d spin orbitals were divided into Ce 3d_{3/2} and Ce 3d_{5/2}, in which the "V line" represents Ce³⁺ with four peaks (V⁰, V¹, V² and V³), and the "U line" represents Ce⁴⁺ with six peaks (U⁰, U¹, U², U³, U⁴ and U⁵). Our findings indicate the incremental content of Ce³⁺ with the proceeding of light irradiation. We have included the results and brief discussion in the revised manuscript and Supplementary Fig. 36a, also shown below:

Supplementary Fig. 36 | **a** In situ Ni 2p XPS spectra for NF@0.1%Ni@CeO₂-V_o. **b** In situ Ce 3d XPS spectra for NF@0.1%Ni@CeO₂-V_o.

Supplementary Fig. 30 | a Normalized X-ray absorption near-edge spectra at the Ni K-edge.

...In situ electron paramagnetic resonance (EPR) also provided compelling evidence for the increase in V_o in the catalysts under concentrated solar irradiation conditions (Fig. 2h), which enhances the recombination between electrons and holes. Photogenerated carriers can be captured by Ni single atoms and release high-frequency phonons *via* recombination. Meanwhile, Ni single atoms and V_o form atomic-scale active sites. These active sites, combined with hot spot effect induced by carriers' recombination, promote the CO_2 reduction reaction. The surface temperature elevation with the introduction of Ni single atoms and V_o is shown in Supplementary Fig. 35a. Besides, V_o indeed possess the capability to capture photo-generated charges (Supplementary Figs. 35b, c). On the other hand, the in situ XPS spectrum demonstrates the valence state change of Ni was undetectable due to its low content of 0.1% (Detailed description in Supplementary Fig. 36a). Conversely, the content of Ce^{3+} increased with light irradiation. This result also suggests the increased V_o content under concentrated solar irradiation (Supplementary Fig. 36b).

Supplementary Discussion: In-situ XPS spectra of the Ni 2p orbitals of the NF@0.1%Ni@CeO₂-V_o catalyst after irradiation with the reactants CO₂ and H₂O vapour for 120 min. However, the spectra only show the satellite peak at 881.04 eV. The signal intensity of Ni is very weak due to its low content (0.1%) (Supplementary Fig. 36a).

Reviewer #3:

“1. In this manuscript, the authors report a single-atom Ni anchored oxygen-vacancy (V_o) rich CeO_2 catalyst to achieve a CH_4 yield of $385.5 \mu\text{mol}/\text{cm}^2$ with an energy conversion efficiency of 1.14% and a selectivity $\sim 100\%$ from photothermal CO_2 reduction in H_2O vapour under concentrated light irradiation. A lot of characterizations and theoretical calculations have been performed, and some results are interesting. However, the organization and writing of the manuscript is poor in whole, lacking a main line logically all through the manuscript. Following are some questions and comments.”

We are grateful to the reviewer for the overall positive feedback on our work and for the valuable suggestion and comments. We have diligently addressed each of these comments in a point-by-point manner and have incorporated additional contents into the revised manuscript.

“2. As the authors mentioned (lines 71-73), CeO_2 has abundant V_o and lattice defects on its surface intrinsically. How did the authors obtain the $NF@0.1\%Ni@CeO_2$ (less or no V_o) as a counterpart of $NF@0.1\%Ni@CeO_2-V_o$?”

We are grateful for this valuable comment. In this work, monatomic Ni and oxygen defects can be created by adding a sodium borohydride reducing agent at high temperature and pressure, as described in Method section 1.2.2. The reducing agent reacts with the lattice oxygen on the surface of the metal oxide during the hydrothermal process to create oxygen vacancies (V_o). Tightly regulating the hydrothermal temperature can influence the chemical reaction rate and thermodynamic environment during catalyst nucleation, thereby controlling the V_o concentration. Therefore, hydrothermal temperature is a crucial factor in the formation of V_o . It has been reported that an increase in temperature can promote the formation of V_o during the hydrothermal synthesis of oxide materials (*Appl. Surf. Sci.*, 2015, **351**, 164–168; *L. Ind. Eng. Chem. Res.*, 2020, **59**, 118–128; *J. Nanobiotechnol.*, 2022, **20**, 507). To provide further evidence, we tested the electron paramagnetic resonance (EPR) of samples at different temperatures and measured the V_o concentration (Supplementary Fig. 22). The experiment revealed that CeO_2 prepared at 110 °C does not form V_o . However, increasing the temperature

leads to a higher concentration of V_o . We have included the results and brief discussion in the revised manuscript, also shown below:

Supplementary Fig. 22 | EPR spectra of catalysts with different hydrothermal synthesis temperatures.

...Additionally, the optimal catalyst preparation conditions were screened (Supplementary Figs. 20, 21). The concentration of V_o can be regulated by the hydrothermal temperature (Supplementary Fig. 22)...

“3. While extensive XAS/EXAFS characterizations and DFT calculations are conducted, the coordination of Ni single atoms in CeO_2-V_o are not described clearly. In abstract, the authors claimed that “the single atoms Ni are precisely anchored on the surface V_o sites of CeO_2 ”; In the EXAFS characterization (lines 106, 107, “These results suggest that the introduction of V_o causes surface reconstruction, with Ni replacing Ce and selectively occupying the V_o ,”; In lines 110-111, the density functional theory (DFT) (Figs. 1j, k) revealed that the single atom Ni loaded on the (111) face by replacing Ce, forming two structures with coordination numbers and bond lengths that align with the XAS results (Supplementary Table 3).”

--- What is the true case? If Ni replaces Ce, can it be taken as doping CeO_2 with Ni simply?”

We sincerely appreciate the efforts made by the reviewer in carefully reviewing our manuscript and providing valuable feedback. We are also very sorry for making misleading. **The actual situation is that Ni single atoms replaces Ce around the surface V_o sites.**

Basically, single-atom catalysts can be classified as either loaded or embedded (*Adv. Mater.*, 2023, 35, 2210575). The catalyst synthesized in this paper belongs to the latter category. However, unlike conventional doping, which involves introducing dopant elements during substrate material synthesis and distributing them throughout both the bulk phase and surface, this study constructs surface defects after substrate material synthesis and introduces monatomic co-catalysts simultaneously to form active sites with a well-defined atomic coordination structure. In conclusion, embedded monatomic doping is a precise surface doping method. The statement has been revised based on the comments below:

...Herein, we report an oxygen-vacancy (V_o) rich CeO_2 catalyst with single-atom Ni anchored around its surface V_o sites by replacing Ce atoms to promote H_2O dissociation and achieve effective photothermal CO_2 reduction under concentrated light irradiation...

...These results suggest that the introduction of V_o causes surface reconstruction, with Ni replacing Ce and anchored around the surface V_o sites, decreasing the H_2O dissociation energy...

“4. (Lines 95-97) “This can be attributed to the presence of V_o , which enables the transfer of electrons from O^{2-} on the catalyst surface to Ni, reducing Ni to the +2 valence state;”
--- How could electrons reduce Ni to Ni^{2+} ? Is it possible? This is wrong in principle.”

We would like to thank the reviewer for proof-reading of our manuscript. We are so sorry to make the typo. The original meaning is the reduction of Ni^{3+} to Ni^{2+} . The sentence has been revised based as also shown below:

...This can be attributed to the presence of V_o , which enables the transfer of electrons from O^{2-} on the catalyst surface to Ni, reducing Ni^{3+} to the +2 valence state...

“5. In Supplementary Fig. 22, the yield of CH_4 increased with temperature, but the yield of CO remained unchanged. Why? How? Please explain.”

We would like to thank the reviewer for this insightful comment. **In fact, as the temperature increases, the yield of CO also increases.** However, due to the significant difference in CH₄ yield, a breakpoint has been added to the ordinate. To avoid misleading, we provide a graph showing the CO yields under different temperature in the inset of Supplementary Fig. 38. We also summarized the product yields of CO and CH₄ in Supplementary Table 11. We have included the changes in the revised manuscript, as also shown below:

Supplementary Fig. 38 | CO₂ reduction performance of the NF@0.1%Ni@CeO₂-V_o catalyst at different temperatures under concentrated solar irradiation conditions. The temperature refers to the ambient temperature of the reactor cavity during catalysis.

Supplementary Table 11 | Variation of CO yields with temperature.

Temperature (°C)	CO Yield (μmol/cm ² /h)	CH ₄ Yield (μmol/cm ² /h)
50	0.15	59.34
100	0.21	79.64
150	0.32	117.41
200	0.48	140.32

...This was confirmed by determining the CH₄ yield at different temperatures under a constant concentrated solar irradiation intensity (Supplementary Figs. 37, 38; Supplementary Tables 10, 11). The CH₄ yield increased with increasing temperature, demonstrating the auxiliary role of heat in the catalysis....

*“6. In Supplementary Fig. 29a, the PL intensity of NF@0.1%Ni@CeO₂-V_o was much lower than that of NF@0.1%Ni@CeO₂. But in Supplementary Fig. 29b, the fitted lifetime (fluorescence decay time) of NF@0.1%Ni@CeO₂-V_o was longer than that of NF@0.1%Ni@CeO₂. This is incorrect conceptually. The authors are suggested to re-check by studying some papers on the fundamentals of photoluminescence (e.g., *Angew, Chem. Int. Ed.* 2023, 62, e202313537, doi.org/10.1002/anie. 202313537). The nonradiative/radiative decays and their respective contributions to catalytic activity should be clarified.”*

We would like to thank the reviewer for the very important comment. We have carefully read the literature and have gained insight into the impact of non-radiative and radiative decay on catalytic activity. In fact, it's not always the case that low fluorescence intensity corresponds to shorter carrier lifetimes. As illustrated in Fig. R2 from the mentioned literature (*Angew, Chem. Int. Ed.* 2023, 62, e202313537), upon adsorption of small molecules on the catalyst surface from air, the fluorescence intensity in air is low, yet the carrier lifetime is long. This phenomenon arises due to the presence of a protective charge, preventing the emission of fluorescence when carriers are captured by the trap state. Therefore, while the overall fluorescence intensity decreases, carrier recombination resulting in fluorescence forms a longer temporal tail. Additionally, this protective charge increases the carrier lifetime by localizing around the trap state and reducing spatial overlap with other charge carriers.

On the most basic level, disregarding the trap state, excited state carriers will undergo either fluorescence-mediated radiative recombination or phonon-mediated non-radiative recombination. Typically, non-radiative recombination tends to dominate the carrier lifetimes. Therefore, a large fluorescence intensity suggests that the non-radiative recombination contributes less to carrier decay, leading to a longer carrier lifetime and a better catalytic performance overall. On the contrary, the existence of trap states introduces a dual effect on carrier dynamics. In the presence of a trap state, there will be a deep trap state-mediated carrier recombination channel, resulting in a reduced percentage of radiative recombination and therefore a lower fluorescence intensity. However, a portion of the carriers captured by the shallow trap state experiences longer lifetime, thereby improving overall carrier lifetime and

catalytic performance (*Joule*, 2022, **6**, 92–133; *J. Am. Chem. Soc.*, 2023, **51**, 28166–28175).

Consequently, the trap states essentially would cause a weaker fluorescence intensity and a longer carrier lifetime. In our study, it is the V_o that acts as the trap state. As a result, their fluorescence intensity decreases and carrier lifetime increases. We therefore include brief discussion in the revised manuscript to explain the phenomenon, as also shown below.

Fig. R2 | **a, b and c** Room-temperature PL spectra of (a) SrTiO₃ and (b) SrTiO₃: Al in air and under vacuum. TRPL decay curves of (c) SrTiO₃: Al in air and under vacuum. The decay curves are fitted with a stretched exponential decay (*Angew. Chem. Int. Ed.*, 2023, **62**, e202313537).

...The synergistic effect of Ni and V_o was found to enhance the electron transport efficiency at the catalyst surface, prolonging the excited state of the fluorescent molecules and thereby substantially reducing the carrier recombination. This phenomenon arises due to the presence of a protective charge, preventing the emission of fluorescence when carriers are captured by the trap state. Therefore, while the overall fluorescence intensity decreases, carrier recombination resulting in fluorescence forms a longer temporal tail. Additionally, this

protective charge increases the carrier lifetime by localizing around the trap state and reducing spatial overlap with other charge carriers. The above analysis explains why V_o acting as a trap state leads to weaker fluorescence intensity yet longer carrier lifetime.

“7. (Lines 150-152) “However, the impurity state of the d orbital is located at the top of the VBM, which implies that the VBM hole capture is equivalent to a relaxation process with an energy level that is similar to the bandgap value between the CBM and VBM, increasing the possibility of capturing the photo-generated holes.”

--- How could an energy level between the CBM and VBM capture the photo-generated holes? Please explain more clearly.”

We would like to thank the reviewer for this insightful comment. To explain why the photo-generated holes could be captured by the energy level between the CBM and VBM, we need to figure out the identities of the CBM/VBM and the energy levels in between. The concept of band gap originates for the bulk phase of the semiconductors, where VBM and CBM are defined. However, when it comes to surface electronic structure, the “levels” between VBM and CBM are not necessarily empty. Theoretically (i.e., in ideal models), the elimination of periodicity at the surface of a semiconductor necessarily introduces states (belongs to surface atoms) between the original bulk phase VBM and CBM (*Stud. Surf. Sci. Catal.*, 1992, **74**, 145–212). For example, V_o on the surface may introduce additional electrons, causing the Fermi level to rise. At this point, the “gap state” may be occupied by electrons. In our study, under the influence of Ni single atoms and V_o , the “gap state” (brought by surface atoms or defects) lies below the Fermi level and is occupied under the non-excited condition. When electrons are excited from the VBM under light irradiation (from bulk phase), an empty state in the VBM is generated. Then the electrons (of surface atoms or defects) in the “gap state” can subsequently transition downwards to the VBM. This is analogous to photo-generated holes transitioning from the VBM to the “gap states” (Fig. 5c and Supplementary 14). Please find below the revision to the detailed description in Supplementary Fig. 14.

Fig. 5 | c Density of states for NF@0.1%Ni@CeO₂-V_o with H₂O adsorption.

Supplementary Fig. 14 | Schematic diagram of photo-generated holes transferred to the gap state.

Supplementary Discussion: In the gap between VBM and CBM, gap states are not necessarily empty. For example, oxygen vacancies may introduce additional electrons, thereby increasing the Fermi level. At this point, the gap states are occupied by electrons. In our system, under the influence of nickel monomers and oxygen vacancies, the gap state lies below the Fermi level and is occupied under the non-excited condition. When electrons are excited from the VBM under light excitation, an empty state in the VBM is generated. Then the electrons of surface atoms or defects in the gap state can subsequently transition downwards to the VBM. This is analogous to photo-generated holes transitioning from the VBM to the gap states.

“8. (Lines 204-207) “In situ electron paramagnetic resonance (EPR) also provided compelling evidence for the increase in V_o in the catalysts under concentrated solar irradiation conditions (Fig. 2h), which enhances the recombination between electrons and holes, thereby increasing the catalyst photothermal conversion efficiency.

--- How could the enhancement of recombination between electrons and holes increase the catalyst photothermal conversion efficiency?”

We would like to thank the reviewer for the helpful comments on our manuscript. This statement is intended to indicate the enhancement of heat utilization efficiency by Ni single atoms and oxygen vacancies (V_o). As illustrated in Fig. 5h, photo-generated carriers can be captured by Ni single atoms and release high-frequency phonons *via* recombination. Meanwhile, Ni single atoms and V_o form atomic-scale active sites. These active sites, combined with hot spot effect induced by carriers' recombination, promote the CO_2 reduction reaction. The surface temperature elevation with the introduction of Ni single atoms and V_o is shown in Supplementary Fig. 35a. Please also find below the revision to the manuscript.

Supplementary Fig. 35 | a Surface temperature of different catalysts under concentrated light conditions.

...In situ electron paramagnetic resonance (EPR) also provided compelling evidence for the increase in V_o in the catalysts under concentrated solar irradiation conditions (Fig. 2h), which

enhances the recombination between electrons and holes. Photogenerated carriers can be captured by Ni single atoms and release high-frequency phonons *via* recombination. Meanwhile, Ni single atoms and V_o form atomic-scale active sites. These active sites, combined with hot spot effect induced by carriers' recombination, promote the CO_2 reduction reaction. The surface temperature elevation with the introduction of Ni single atoms and V_o is shown in Supplementary Fig. 35a...

“9. (Lines 381-383) “The results showed that the C in CO_2 cannot coordinate to the Ni sites of the catalyst surface because they are positively charged. Therefore, CO_2 is not stably adsorbed to the surface by bonding single-atom Ni.” If this is the case, is any other single-atom catalyst not favorable for the adsorption/activation of CO_2 molecules? The fact is that it is not true.”

We would like to thank the reviewer for this pertinent comment. In fact, as shown in Supplementary Figs. 54, 55, if CO_2 is adsorbed on the Ni single-atom *via* a C–Ni bond, the adsorption energy is very low, rendering it ineffective. This is precisely due to the strong electrostatic repulsion between the Ni atom and the C atom. Therefore, the C atom in CO_2 can bond with lattice oxygen, causing a slight deviation of the Ni single atom from its original position and forming a bond with the O atom in CO_2 , resulting in effective adsorption.

However, we also agree with the assertion “it is not true that a single-atom catalyst is not favorable for the adsorption/activation of CO_2 molecules”. In our specific case, it is the preparation method that enables the synthesis of the catalyst containing both Ni single atoms and V_o , along with positive charged Ni sites. Similar instances, where single atoms collaborate with lattice oxygen to facilitate CO_2 adsorption efficacy, can also be found in other literature (*Energy Environ. Sci.* 2022, **15**, 401–868). Please find below the revision to the detailed description in Supplementary Fig. 55.

Supplementary Fig. 54 | Adsorption energies of CO₂ on clean surface of CeO₂ (111), V_o site and Ni SAC site.

Supplementary Fig. 55 | Adsorption structure of CO₂ on clean surface of CeO₂ (111), V_o site and Ni SAC site.

Supplementary Discussion: If CO₂ is adsorbed on the Ni single-atom *via* a C–Ni bond, the adsorption energy is very low, making it ineffective. This is primarily attributed to the strong

electrostatic repulsion between the Ni atom and the C atom. Therefore, the C atom in CO₂ can bond with lattice oxygen, causing a slight deviation of the Ni single atom from its initial position and forming a bond with the O atom in CO₂. Subsequently, this leads to a more efficient adsorption process.

“10. (Lines 412-413) “Any remaining O species may form O₂ molecules with other O species or fill the V_o, leading to Ni deactivation, which could shift from the planar quadrilateral coordination structure to an octahedral one.”

-- Did the authors detect evolution of O₂? Otherwise, Ni deactivation would occur since the O species will fill the V_o, as the authors say.”

We would like to thank the reviewer for this valuable comment. Yes, we have determined the evolution of O₂. Supplementary Fig. 61 displays the reaction rate profile of both O₂ and CH₄ products over a 2-h duration. Additionally, EPR analysis of the catalyst was conducted before and after the reaction, revealing no decrease in the concentration of V_o (Supplementary Fig. 60). The above results indicate that the production of O₂ did not fill the V_o on the catalyst surface. Moreover, the sustained activity of the catalyst throughout prolonged CO₂ reduction reaction suggests that Ni does not become deactivated (Supplementary Fig. 32). These results are involved in the revised manuscript and are also shown below.

Supplementary Fig. 60 | EPR spectra of before and after the catalyst reaction.

Supplementary Fig. 61 | The spectrum of O₂ yield for NF@0.1%Ni@CeO₂-V_o.

Supplementary Fig. 32 | Stability test for photothermal catalysis of CO₂ reduction.

...Any remaining O species may form O₂ molecules with other O species or fill the V_o, leading to Ni deactivation, which could shift from the planar quadrilateral coordination structure to an octahedral one. To confirm the mechanism, we assessed the concentration of oxygen vacancies before and after the reaction. The findings revealed an increase in the concentration of oxygen vacancies after the reaction, thereby implying that the oxygen vacancies were not occupied by O₂ (Supplementary Fig. 60). Moreover, the O₂ yield reached 762.5 μmol/cm² of concentrated solar irradiation (Supplementary Fig. 61)...

“11. In the session “Mechanism of light and thermal effects over the Ni single atom”, the authors intended to prove the photo-thermal coupled CO₂ reduction mechanism. But they did

not design and conduct appropriate comparative experiments as many previously reported research works on photothermal CO₂ reduction did to clarify the respective effects.”

We would like to thank the reviewer for this useful comment. **We have added additional experiments:** 1) Introducing cooling circulating water under concentrated solar irradiation conditions (Supplementary Fig. 16); 2) Measuring the surface temperature of the NF@0.1%Ni@CeO₂-V_o catalyst under UV, Vis, NIR, and full spectrum (Supplementary Fig. 17); 3) Employing the NF@0.1%Ni@CeO₂-V_o catalyst under UV, Vis, NIR, and full-spectrum (Supplementary Fig. 18). These experiments provide evidence for thermal-assisted photocatalysis under concentrated light.

Then, we measured photocurrent (Supplementary Fig. 41) and open circuit potential (OCP) (Supplementary Fig. 43) at different light intensities to investigate the effect of light. Finally, we measured the surface temperatures of different catalysts under UV, Vis, NIR, and full spectrum (Supplementary Fig. 35a) and performed OCP tests at different temperatures (Supplementary Fig. 40) to illustrate the role of thermal effects. These results are involved in the revised manuscript and are also shown below.

Supplementary Fig. 16 | The yield of CH₄ over the NF@0.1%Ni@CeO₂-V_o catalyst under different conditions.

Supplementary Fig. 17 | **a-d** Temperatures of the NF@0.1%Ni@CeO₂-V_o under UV (**a**), Vis (**b**), NIR (**c**) and full-spectrum (**d**) concentrated solar irradiation.

Supplementary Fig. 18 | The yield of products under different wavelength ranges of concentrated light.

Supplementary Fig. 41 | **a** Transient photocurrent curves for different catalysts under concentrated solar irradiation. **b** Transient photocurrent density of NF@0.1%Ni@CeO₂-V_o catalyst under different light intensities.

Supplementary Fig. 43 | Open circuit potential under different light intensities.

Supplementary Fig. 35 | a Surface temperature of different catalysts under concentrated light conditions.

Supplementary Fig. 40 | Open circuit potential at different temperatures.

...Conversely, under simulated solar light irradiation of 1200 mW/cm², the CH₄ yield substantially increased. In addition, catalytic experiments were conducted with cooled circulating water or under irradiation of variable wavelengths (Supplementary Figs. 16, 17, 18). The catalytic reaction is facilitated by raising the temperature and photogenerated electrons play a key role. The above results suggest that the CO₂ photo-thermal catalysis is proceeded in a manner of thermally assisted photo-catalysis....

...Meanwhile, the photo-response of NF@0.1%Ni@CeO₂-V_o was the most substantial under concentrated solar irradiation conditions (Supplementary Figs. 41, 42a, b), suggesting that the high photo-electron flux density drives single-atom Ni and V_o to further enhance the conductivity and carrier separation efficiency. Furthermore, the photo-generated electrons under concentrated solar irradiation not only increase in quantity but also exhibit enhanced quality, which can be attributed to the increase in the Fermi energy level in NF@0.1%Ni@CeO₂-V_o (Supplementary Fig. 43)...

...In situ electron paramagnetic resonance (EPR) also provided compelling evidence for the increase in V_o in the catalysts under concentrated solar irradiation conditions (Fig. 2h), which enhances the recombination between electrons and holes. Photogenerated carriers can be captured by Ni single atoms and release high-frequency phonons *via* recombination. Meanwhile, Ni single atoms and V_o form atomic-scale active sites. These active sites, combined with hot spot effect induced by carriers' recombination, promote the CO₂ reduction reaction. The surface temperature elevation with the introduction of Ni single atoms and V_o is shown in Supplementary Fig. 35a.

...However, because the migration rate is greater than the complexation rate, heat is favourable for the reaction. OCP tests at variable temperatures also validate this mechanism (Supplementary Fig. 40)...

“12. Regarding the roles of Ni single atom and V_o , the authors seem not to give consistent descriptions that are very confusing to the reviewer. 1) “The CO_2 temperature-programmed desorption spectra shown in Supplementary Fig. 28 confirm that V_o and Ni can provide surface basic sites to facilitate the CO_2 adsorption and decomposition.” 2) The in situ DRIFTS show that “The remarkably strong $m-CO_3^{2-}$ peaks indicate that CO_2 is adsorbed on the lattice O of CeO_2 instead of on Ni.”

--- What is the true case? Please give more explanations.”

We would like to thank this reviewer for the proof-reading of our manuscript. We are sorry for the unsatisfying statement on the CO_2 reaction sites. In fact, as aforementioned, lattice oxygen in CeO_2 serves as the sites for adsorbing CO_2 molecules by bonding with C atom from CO_2 . This adsorption, together with the surrounding V_o structure, makes Ni single atoms slightly deviated from their original positions and further bonded with O atoms in CO_2 (see Supplementary Figs. 54, 55). Therefore, both the mentioned sentences are provided the right meaning of the true mechanism. Please find below the revision to the detailed description in Supplementary Fig. 55.

Supplementary Fig. 54 | Adsorption energies of CO_2 on clean surface of CeO_2 (111), V_o site and Ni SAC site.

Supplementary Fig. 55 | Adsorption structure of CO₂ on clean surface of CeO₂ (111), V_o site and Ni SAC site.

Supplementary Discussion: If CO₂ is adsorbed on the Ni single-atom *via* a C–Ni bond, the adsorption energy is very low, making it ineffective. This is primarily attributed to the strong electrostatic repulsion between the Ni atom and the C atom. Therefore, the C atom in CO₂ can bond with lattice oxygen, causing a slight deviation of the Ni single atom from its initial position and forming a bond with the O atom in CO₂. Subsequently, this leads to a more efficient adsorption process.

“13. Some experimental details. 1) The light source spectra and intensities of both the concentrated solar irradiation and the non-concentrated solar irradiation should be provided. 2) The isotope test should be performed to confirm the origin of CH₄ is from CO₂ but not from other possible carbon-containing sources and impurities. 3) The authors used the unit of $\mu\text{mmol cm}^{-2}$ or $\mu\text{mmol cm}^{-2} \text{h}^{-1}$ for evaluating the catalytic activity. However, most previously

reported works used the unit of $\mu\text{mmol g}_{\text{cat}}^{-1} \text{h}^{-1}$. In the Supplementary table 5, the comparison of the authors' catalyst with others would not be appropriate.”

We would like to thank the reviewer for the very important comments and suggestion.

Firstly, the source spectrum and intensity of non-concentrated solar irradiation was measured using a fibre-optic spectrometer, as shown in Supplementary Fig. 13. However, for the measurement of concentrated solar irradiation, the intensity is out of the detecting range of the fibre-optic spectrometer. As a result, we cannot obtain its spectrum and intensity information. In principle, the light intensity is only changed by the Fennel lens and the attenuator, and has no effect on the spectrum. The multiple for concentrate light is the same as the multiple for light intensity. The results indicate a value of 420 mW/cm^2 for non-concentrated irradiation and 4200 mW/cm^2 for concentrated irradiation.

Supplementary Fig. 13 | Light source spectra of non-concentrated solar irradiation.

...The infrared (IR) images presented in **Supplementary Fig. 11** indicate the surface temperatures of $\text{NF@0.1\%Ni@CeO}_2\text{-V}_o$ and NF@0.1\%Ni@CeO_2 under concentrated solar irradiation (those recorded under non-concentrated solar irradiation conditions are presented in **Supplementary Fig. 12**). Meanwhile, light source spectra are provided in **Supplementary Fig. 13**...

Secondly, $^{13}\text{CO}_2$ isotope labeling experiments were carried out under the same conditions. The gas products were tested by gas chromatography-mass spectrometry (GC-MS, Agilent

8860). In the experiment, helium gas was used as a carrier gas, and sample injection temperature and auxiliary heating temperature were both set at 125 °C, with the constant flow rate being 1.2 mL·min⁻¹. The initial temperature of the column box is 60 °C, and the temperature is heated to 180 °C at 10 °C·min⁻¹, and the storage temperature is kept for 1 min. As shown in Supplementary Fig. 24, the products were detected by gas chromatography-mass spectrometry (GC-MS). The main signal at m/z = 17 is assigned to ¹³CH₄ and the other signal at m/z = 29 can be assigned to ¹³CO. This result indicates that the CO and CH₄ generation originates from the CO₂ reactant, thereby ruling out catalyst decomposition or carbon pollution as contributing factors. Please also find below the revision.

Supplementary Fig. 24 | ¹³CO₂ isotopic tracing experiments of NF@0.1%Ni@CeO₂-V_o.

...To identify the carbon source of the reduced products, isotope tracing experiments were carried out using reactant ¹³CO₂. The main signal at m/z = 17 is assigned to ¹³CH₄ and the other signal at m/z = 29 can be assigned to ¹³CO. This result indicates that the CO and CH₄ generation originates from the CO₂ reactant, thereby ruling out catalyst decomposition or carbon pollution as contributing factors (Supplementary Fig. 24).

Thirdly, the performance of catalysts is indeed crucial, but the units used to evaluate catalyst performance are diverse. Yes, commonly, the unit in the field of CO₂-H₂O photocatalysis is μmol/g/h. However, this work utilises a fixed-bed reactor with the catalyst fixed in a supported substrate material. It is important to consider the area of the catalyst that is exposed to light. There are related literature reports using μmol/cm² as the unit (*Nature*, 2021, **598**, 304–307;

Joule, 2018, 2, 2667–2680). In addition, direct comparison reaction rates of different catalysts are not meaningful due to differences in reactors, illumination mode, fixed-bed, fluidised-bed, light intensity, irradiated area, etc. Therefore, it is most reasonable to compare energy conversion efficiencies (Supplementary Table 5).

Supplementary Table 5 | Comparison of energy conversion efficiencies of photo and photothermal catalysis.

Catalysts	Light Intensity (mW cm ⁻²)	Reaction medium	Reduction products	Catalyst dosage (mg)	System temperature (K)	Yield	η _{STC} (%)
NF@0.1%Ni@CeO ₂ -V _o (This work)	4200	CO ₂ and H ₂ O vapour	CH ₄	1	635.15	192.75 μmol cm ⁻² h ⁻¹	1.14
WO ₃	4000	CO ₂ and H ₂ O vapour	CO and CH ₄	1	511.15	26.51 and 3.36 μmol·cm ⁻² ·h ⁻¹	0.073
ZrO ₂	4000	CO ₂ and H ₂ O vapour	CO and CH ₄	1	529.15	20.63 and 7.36 μmol·cm ⁻² ·h ⁻¹	0.086
TiO ₂	4000	CO ₂ and H ₂ O vapour	CO and CH ₄	1	485.15	15.42 and 2.65 μmol·cm ⁻² ·h ⁻¹	0.047
Au-Mt-H6 ²	720	CO ₂ and H ₂ O vapour	CO and CH ₄	10	454.15	0.14 and 0.17 μmol m ⁻²	0.000037
TF@TNT/0.4CoO _x -0.1CuO ₃	4266	CO ₂ and H ₂ O vapour	CO and CH ₄	1	573.15	8.45 and 58.2 μmol cm ⁻² h ⁻¹	0.35
Bi ₂ S ₃ @In ₂ S ₃ ⁴	1150	CO ₂ and H ₂ O vapour	CO and C ₂ H ₄	5	513.15	1.95 and 11.81 μmol g ⁻¹ h ⁻¹	0.00033

Pt/ZnO ⁵	4200	CO ₂ and H ₂ O vapour	CO and CH ₄	20	540.15	115.32 and 8 μmol g ⁻¹ h ⁻¹	0.264
Au _x /TJU-16 ⁶	300 W Xe lamp	CO ₂ and H ₂ O vapour	CO and CH ₄	40	/	10.1 and 2.5 μmol g ⁻¹ h ⁻¹	0.034
Au@Cr ₂ O ₃ ⁷	1600	CO ₂ and H ₂ O vapour	CO and H ₂	1000	/	0.074 and 0.013 mol g ⁻¹ h ⁻¹	0.89
CuO _x /TiO ₂ ⁸	300 W Xe lamp	CO ₂ and H ₂ O vapour	CO and CH ₄	50	/	0.32 and 2.29 μmol g ⁻¹ h ⁻¹	0.0372
Sn/In ₂ O ₃ ⁹	300 W Xe lamp	CO ₂ and H ₂ O vapour	CO	50	/	17.9 μmol g ⁻¹ h ⁻¹	0.05
Rh/Al nanoantenna ¹⁰	11300	CO ₂ and H ₂	CH ₄ and H ₂ O	20	973.15	550 mmol g ⁻¹ h ⁻¹	0.0011
Co/Al ₂ O ₃ ¹¹	3295	CO ₂ and H ₂	CH ₄	/	803.15	6036 μmol g ⁻¹ h ⁻¹	0.0002

“14. The reviewer’s personal taste. Using concentrated solar irradiation to improve the catalytic efficiency is more likely a technical skill but not a scientific merit. Given the aforementioned questions and comments, I could not recommend acceptance of the current paper.”

We genuinely thank the reviewer for the comments, while are also very sorry about his/her concern on the scientific merit of the research.

In fact, as far as we can see, there are still many scientific questions unsolved enclosing photocatalytic CO₂ reduction over years. With the deepening of the understanding on this system, the microscale mechanisms, especially those about the behavior of physical basis particles, become increasingly essential.

To date, most of current studies remain narrow in focus dealing only with the inertia carrier kinetics or chemical reactions, while overlooked the importance of high input

photon flux, which is probable to induce brand new micro mechanisms. Just in very recent years, the concentrated sunlight technology has begun to be utilized in photothermal reduction of CO₂. The improvement alongside the introduce of concentrated solar irradiation brought researchers new scientific questions, including but not limited to photon-phonon conversion, photon-electron interaction, electron-phonon coupling effect under high photon flux input. Among them, photon-phonon conversion exerts heats losses in traditional photocatalytic CO₂ reduction, while plays a crucial role in photothermal effects under concentrated light irradiation. Photon-electron interaction was regarded as a constant energy input in traditional photocatalytic CO₂ reduction, while it was found to elevate the average electron energy (by raising fermi level) and thereby promoting chemical reaction under concentrated light irradiation. Electron-phonon coupling effect was acknowledged to generally having negative effect on carrier's lifetime, and under concentrated light irradiation, it becomes a much more critical issue due to the notable temperature elevation. Besides, there are hot electrons, local surface plasmonic resonance (LSPR) effect, stability challenge serving as challenging scientific questions enclosing photocatalytic CO₂ reduction under concentrated solar irradiation. Therefore, we believe this system with scientific merits deserve further development.

Furthermore, many research groups have already utilized concentrated solar irradiation to improve the catalytic efficiency, yet failed to comprehensively figure out the underlying micro-mechanisms. Zetian Mi' team have developed a strategy to achieve a high STH efficiency of 9.2% using pure water, concentrated solar light and an indium gallium nitride photocatalyst. A temperature-controllable photocatalytic system was designed to perform Electrolytic hydrogen production technology (OWS) in pure water at different temperatures as shown in Fig. R3a (*Nature*, 2023, **613**, 66–70). As for lab-scale concentrated photocatalysis system, Hanfeng Lu's team provided a scheme of the home-made concentrating solar light reactor system as illustrated in Fig. R3b (*ACS Appl. Energy Mater.*, 2019, **2**, 8376–8380). Clemens Suter's team presented the successful scaling of a thermally integrated photoelectrochemical device utilizing concentrated solar irradiation to a kW-scale pilot plant capable of co-generation of hydrogen and heat. The solar energy to the hydrogen, oxygen and heat co-generation system and an instrumentation diagram of material and energy flows have shown in Fig. R3c and R3d (*Nat Energy*, 2023, **8**, 586–596). However, above research still

cannot explain why the super-linear energy conversion efficiency occurs. Therefore, it can be inferred that more research is needed to investigate the driven force that leads to the extraordinary performance of photocatalysis under concentrated solar irradiation.

In summary, photocatalysis under concentrated solar irradiation has presented extraordinary performance yet the underlying mechanism remains unclear. We primarily elucidate the intrinsic reasons behind the research system's high catalytic efficiency and employ a variety of research methodologies in the investigation of the microscopic mechanism, endeavoring to provide explanations with the utmost rigor. Our work proposed a synergistic mechanism induced by the V_o defects coordinated with single atom Ni, which significantly promote the capture of charge carriers and local phonons at the Ni d-impurity state orbitals, thereby inducing more effective activation of H_2O . This strategy effectively resolves the challenging issue of H_2O dissociation, which has been a limiting factor in the efficiency of STC energy conversion. We believe that this system possesses significant scientific value, and the elucidation of its underlying mechanisms holds long-term implications for the efficient production of hydrocarbon fuels based on solar energy.

Fig. R3 | a STH efficiency of Rh/Cr₂O₃/Co₃O₄-loaded InGaN/GaN NWs in tap water and sea water, with outdoor photocatalytic OWS system system on North Campus at the University of Michigan, Ann Arbor, Michigan, United States. **b** Scheme of the home-made concentrating

solar light reactor system. **c** Technical illustration of the overall site showing key components such as the solar parabolic concentrator dish, reactor and ancillary hardware and cabinets and **d** a simplified process and instrumentation diagram of the system showing material and energy flows.

REVIEWERS' COMMENTS

Reviewer #1 (Remarks to the Author):

All concerns have been well addressed and I recommend publication of this manuscript in current form.

Reviewer #2 (Remarks to the Author):

The authors have properly addressed my previous concerns in the revised version of the manuscript. Therefore, the current manuscript is recommended for publication by the journal.

Reviewer #3 (Remarks to the Author):

It is obvious that the authors have made great efforts in modifying the manuscript. Essentially, all my questions, comments, and concerns have been properly addressed in the revised manuscript. Thus, I would like to recommend acceptance of the revised manuscript.

Point-by-Point Responses to Reviewers' Comments

Reviewer #1:

“All concerns have been well addressed and I recommend publication of this manuscript in current form.”

We are grateful for the thoughtful comments and recommendations provided by this reviewer. Thank you for your valuable input.

Reviewer #2:

“The authors have properly addressed my previous concerns in the revised version of the manuscript. Therefore, the current manuscript is recommended for publication by the journal.”

We are grateful for the thoughtful comments and recommendations provided by this reviewer. Thank you for your valuable input.

Reviewer #3:

“It is obvious that the authors have made great efforts in modifying the manuscript. Essentially, all my questions, comments, and concerns have been properly addressed in the revised manuscript. Thus, I would like to recommend acceptance of the revised manuscript.”

We are grateful for the thoughtful comments and recommendations provided by this reviewer. Thank you for your valuable input.